# SERBP1 interacts with PARP1 and is present in PARylation-dependent protein complexes regulating splicing, cell division, and ribosome biogenesis

Kira Breunig[1†], Xuifen Lei[1†], Mauro Montalbano[2,3†], Gabriela DA Guardia[4], Shiva Ostadrahimi[1,5], Victoria Alers[1,5,6], Adam Kosti[1,5], Jennifer Chiou[7], Nicole Klein[1], Corina Vinarov[1], Lily Wang[1], Mujia Li[1], Weidan Song[8], W Lee Kraus[8], David S Libich[1,6], Stefano Tiziani[7,9,10], Susan T Weintraub[6], Pedro AF Galante[4], Luiz O Penalva[1,5*]

[1]Children's Cancer Research Institute, UT Health San Antonio, San Antonio, United States; [2]Mitchell Center for Neurodegenerative Diseases, University of Texas Medical Branch, Galveston, United States; [3]Department of Neurology, University of Texas Medical Branch, Galveston, United States; [4]Centro de Oncologia Molecular, Hospital Sírio-Libanês, São Paulo, Brazil; [5]Department of Cell Systems and Anatomy, UT Health San Antonio, San Antonio, United States; [6]Department of Biochemistry and Structural Biology, UT Health San Antonio, San Antonio, United States; [7]Department of Nutritional Sciences, College of Natural Sciences, University of Texas at Austin, Austin, United States; [8]Laboratory of Signaling and Gene Regulation, Cecil H. and Ida Green Center for Reproductive Biology Sciences, The University of Texas Southwestern Medical Center, Dallas, United States; [9]Department of Pediatrics, Dell Medical School, University of Texas at Austin, Austin, United States; [10]Department of Oncology, Dell Medical School, University of Texas at Austin, Austin, United States

*For correspondence: penalva@uthscsa.edu

[†]These authors contributed equally to this work

Competing interest: The authors declare that no competing interests exist.

## eLife Assessment

This study reports **valuable** insights into the interactome of the RNA-binding protein SERBP1 and possible links through PARylation to diverse processes, including splicing, cell division, and ribosome biogenesis. The diversity of processes SERBP1 may regulate means this work would be of very broad interest to the cell biology community. The proteomics data are **solid**, but the functional connection to downstream processes and the link to Alzheimer's disease, while **compelling**, still require further examination. These latter data currently rely on a very limited set of experiments and patient samples with questionable quality of preservation and methodology.

**Abstract** RNA binding proteins (RBPs) containing intrinsically disordered regions (IDRs) are present in diverse molecular complexes where they function as dynamic regulators. Their characteristics promote liquid-liquid phase separation (LLPS) and the formation of membraneless organelles such as stress granules and nucleoli. IDR-RBPs are particularly relevant in the nervous system and their dysfunction is associated with neurodegenerative diseases and brain tumor development. Serpine1 mRNA-binding protein 1 (SERBP1) is a unique member of this group, being mostly disordered and lacking canonical RNA-binding domains. We defined SERBP1's interactome, uncovered novel roles in splicing, cell division and ribosomal biogenesis, and showed its participation

in pathological stress granules and Tau aggregates in Alzheimer's brains. SERBP1 preferentially interacts with other G-quadruplex (G4) binders, implicated in different stages of gene expression, suggesting that G4 binding is a critical component of SERBP1 function in different settings. Similarly, we identified important associations between SERBP1 and PARP1/polyADP-ribosylation (PARylation). SERBP1 interacts with PARP1 and its associated factors and influences PARylation. Moreover, protein complexes in which SERBP1 participates contain mostly PARylated proteins and PAR binders. Based on these results, we propose a feedback regulatory model in which SERBP1 influences PARP1 function and PARylation, while PARylation modulates SERBP1 functions and participation in regulatory complexes.

## Introduction

RNA binding proteins (RBPs) are a diverse group of regulators with 1500 members cataloged in the human genome (*Gerstberger et al., 2014*; *Corley et al., 2020*). They display a variety of RNA binding domains (RRMs, zinc-fingers, KH domains, dsRBD, etc.) and control multiple stages of gene expression, including RNA processing (splicing, capping, 3' end processing, and polyadenylation), modification, transport, decay, localization, and translation (*Corley et al., 2020*; *Kelaini et al., 2021*). RBPs are particularly relevant in the nervous system, where they regulate critical aspects of neurogenesis, neuronal function, and nervous system development (*Chan et al., 2022*; *Bryant and Yazdani, 2016*). Alterations in RBP expression and function are linked to neurological disorders, neurodegenerative diseases, and brain tumor development (*Prashad and Gopal, 2021*; *Marcelino Meliso et al., 2017*; *Sanya et al., 2023*; *Kavanagh et al., 2022*; *Velasco et al., 2019*; *Gebauer et al., 2021*; *Nussbacher et al., 2015*).

RBPs assemble in complexes to regulate gene expression. RBP interactions can take place in the context of macromolecular complexes associated with liquid-liquid phase separation (LLPS). This process supports important regulatory environments and is the foundation for membraneless organelles such as stress granules, Cajal bodies, nucleoli, and P-bodies (*An et al., 2021*; *Drino and Schaefer, 2018*; *Gomes and Shorter, 2019*). RBPs in LLPS tend to share characteristics that include intrinsically disordered regions (IDRs) and RGG boxes. IDR-RBPs can interact with RNA without the presence of specific folded RNA-binding domains (*Zeke et al., 2022*). These features provide unique structural and biophysical characteristics to dynamically regulate translation, stress response, RNA processing, and neuronal function (*Roden and Gladfelter, 2021*).

Many RBPs in LLPS share a binding preference for G-quadruplexes (G4s). G4s are complex DNA or RNA structures that display stacked tetrads of guanosines stabilized by Hoogsteen base pairing. They function as regulatory elements in different stages of gene expression, promote LLPS, and are implicated in cancer, viral replication, neurodegenerative diseases, neurological disorders, and prion diseases (*Cave and Willis, 2022*; *Masai and Tanaka, 2020*; *Nakanishi and Seimiya, 2020*; *Kosiol et al., 2021*).

RBPs containing IDRs and RGG motifs are particularly relevant in the nervous system. Their misfolding contributes to the formation of pathological protein aggregates in Alzheimer's disease (AD), Frontotemporal Lobar Degeneration (FTLD), Amyotrophic Lateral Sclerosis (ALS), and Parkinson's disease (PD) while their aberrant expression has been linked to glioblastoma development (*Darling and Shorter, 2021*; *Webber et al., 2020*; *Uversky, 2017*). SERBP1 (Serpine1 mRNA-binding protein 1) is a unique member of this group of RBPs for being mostly disordered and lacking typical RNA binding domains other than two short RG/RGG motifs (*Baudin et al., 2021*). SERBP1 is a highly conserved protein with homologs identified in yeast, plants, invertebrates, and vertebrates. SERBP1 is highly expressed in neuronal and glioma stem cells and a decrease in its levels is required for neuronal differentiation (*Kosti et al., 2020*). In glioblastoma, SERBP1 functions as a central regulator of metabolic pathways, coordinating the expression of related enzymes and associated factors implicated in serine biosynthesis, one-carbon, and 5'-methylthioadenosine (MTA) cycles. Ultimately, SERBP1 impacts methionine production and, consequently histone methylation and the expression of genes implicated in neuronal differentiation (*Kosti et al., 2020*).

In the present study, we investigated the SERBP1 interactome. Overall, our results established SERBP1 as a multi-regulatory protein, corroborating that RBPs containing IDRs are dynamic and can assemble different complexes. Despite its predominant cytoplasmic localization, SERBP1 is associated

with proteins in specific nuclear complexes implicated in splicing regulation, cell division, and ribosome biogenesis. Approximately a third of SERBP1-associated proteins bind to G4s. These proteins are implicated in various stages of gene expression, indicating that G4s function as SERBP1 regulatory motifs in different scenarios. Finally, our study has identified important associations between SERBP1 and PARP1 that suggest a regulatory loop, in which SERBP1 influences PARylation, and PARP activity. At the same time, PARylation of SERBP1 and its partner proteins modulate their association and function.

## Results

### SERBP1 interactome reveals its associations with diverse regulatory complexes

To identify SERBP1-associated proteins and molecular complexes in which it is involved, we conducted pull down experiments in 293T cells followed by proteomics analysis. Cells were transfected with pSBP-SERBP1 or pUltra-SERBP1(control). The SBP tag is a short peptide with high affinity for streptavidin. Protein complexes are then isolated via pulldown using streptavidin beads. Analysis by mass spectrometry yielded a high-confidence list of 570 SERBP1-associated proteins. Peptides for these proteins were only present in the pSBP-SERBP1 pull-down samples or displayed a minimum of three-fold enrichment in comparison to control – *Supplementary file 1*, *Figure 1—figure supplement 1*. Gene Ontology (GO) analyses (*Ge et al., 2020*; *Zhou et al., 2019*; *Supek et al., 2011*) of identified partners indicated that SERBP1 participates in diverse regulatory complexes in the cytoplasm and nucleus – *Figure 1A and B* and *Supplementary file 1*. We compared our data to the results of two large-scale studies that used proximity-dependent biotinylation (Bio ID; *Go et al., 2021*; *Youn et al., 2018*) to identify protein-protein interactions and checked the datasets in BioGRID (*Stark et al., 2006*). 146 SERBP1 interactors identified in our analysis were also identified in other high-throughput studies – *Supplementary file 1*.

Although SERBP1's partners indicate its participation in different stages of translation as well as in its regulation, the many associations with translation initiation factors suggest that SERBP1's primary function is in initiation – *Supplementary file 1*. As expected, based on previous studies (*Brown et al., 2018*; *Muto et al., 2018*; *Ahn et al., 2015*; *Martini et al., 2021*), we identified several ribosomal proteins associated with SERBP1 – *Supplementary file 1*. SERBP1 knockdown in U251 and U343 cells resulted in a decrease of overall translation as shown by puromycin incorporation assays – *Figure 1—figure supplement 2*.

SERBP1 is mostly localized in the cytoplasm. SERBP1 nuclear localization increases in response to stress and is regulated by arginine methyltransferases (*Lee et al., 2014*; *Lee et al., 2012*; *Passos et al., 2006*). SERBP1-associated proteins suggest it participates in nuclear functions; out of 570 associated proteins, 344 display cytoplasmic/nuclear localization, while 100 are exclusively found in the nucleus – *Supplementary file 1*. GO analyses of this group identified specific nuclear complexes. The functions of SERBP1-associated proteins suggest that SERBP1 is involved in different stages of ribosome biogenesis – *Supplementary file 1*. SERBP1 participation in ribosome biogenesis is corroborated by its presence in nucleoli – *Figure 1—figure supplement 2B, C*. Results of our previous RIP-Seq analysis (*Kosti et al., 2020*) and a CLIP-seq study (*Martini et al., 2021*) showed that SERBP1 binds preferentially to multiple snoRNAs (*Supplementary file 2*) and at specific sites in 18 S and 28 S rRNA. These results, aligned with the fact that SERBP1 associates with FBL and DKC1, two critical snoRNP components, strongly indicate SERBP1 involvement in rRNA modification. Besides ribosome biogenesis, SERBP1 is also likely involved in different stages of RNA processing, in particular splicing, and telomere maintenance – *Figure 1A and B*.

SERBP1 is a highly conserved protein. In *S. cerevisiae*, SERBP1 homolog is STM1 (*Coppolecchia et al., 1993*). According to BioGRID (*Stark et al., 2006*), 183 proteins were identified as STM1 interactors. Of the 144 identified human homologs, we found that 53 were SERBP1 interactors – *Supplementary file 3*. GO analyses of STM1 interactors indicated that STM1 is also a multi-functional protein. Comparisons of enriched GO terms from the SERBP1 and STM1 analyses suggest that several regulatory functions assigned to SERBP1 based on its associated proteins are evolutionarily conserved, including translation, ribosome biogenesis, RNA processing and telomere maintenance – *Supplementary file 3*. SERBP1 also has homologs in plants. A recent study in *A. thaliana* established that AtRGG

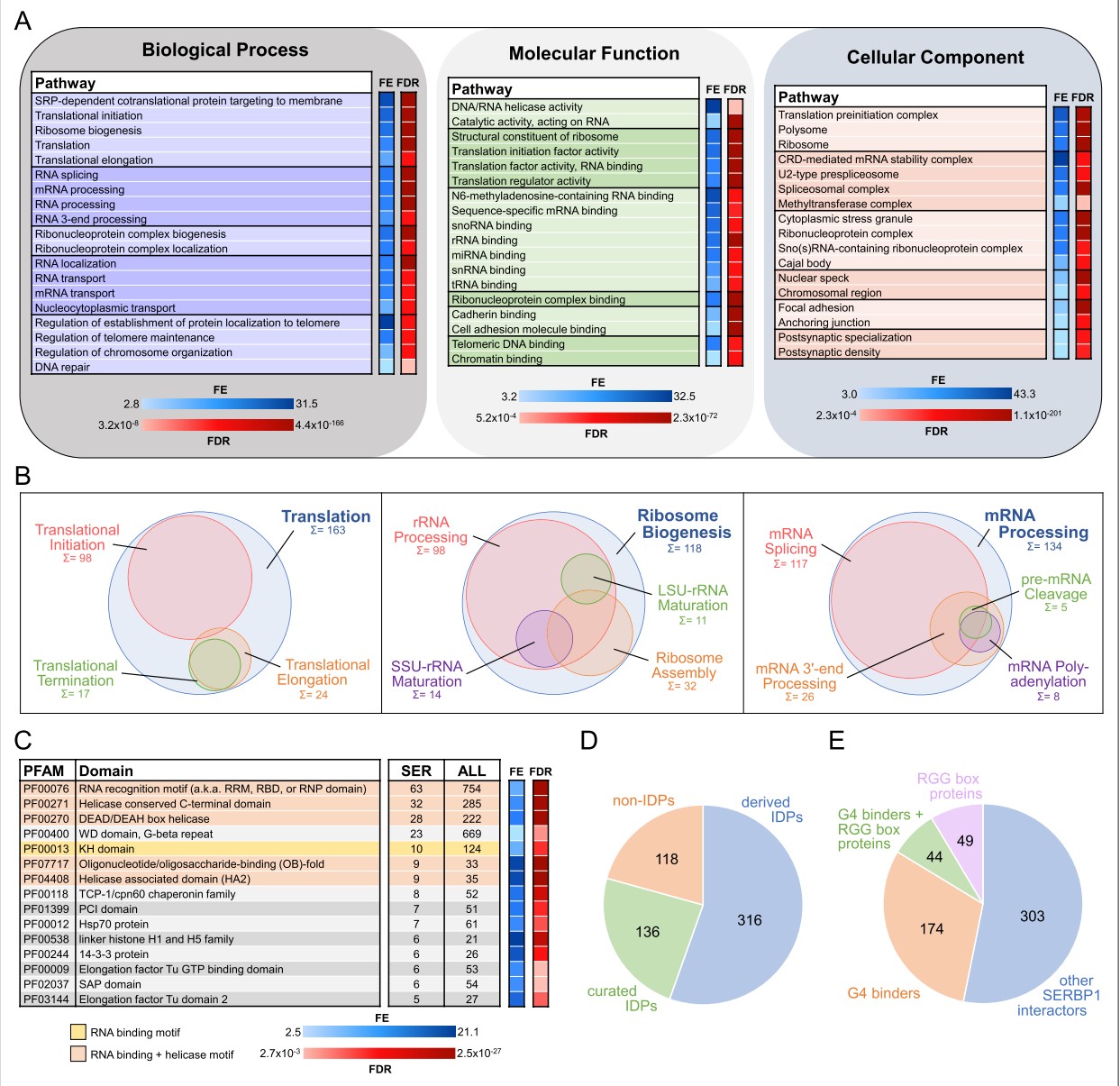

**Figure 1.** Gene ontology (GO) analysis and characterization of SERBP1-interacting proteins. (**A**) SERBP1-interacting factors were identified via pull-down analysis in 293T cells. GO enrichment analysis (Biological Processes, Molecular Function, and Cellular Component) was performed with ShinyGO (*Ge et al., 2020*). A selection of the most relevant GO terms is represented. FE = fold enrichment; FDR = false discovery rate. (**B**) Visualization of associated GO-enriched terms in three main regulatory branches: translation, ribosome biogenesis, and RNA processing. Sum and circle size represent the count of SERBP1 interactors assigned to each GO term. (**C**) Representation of top enriched protein domains according to PFAM (*Sonnhammer et al., 1997*) among SERBP1 interactors. Only PFAM domains with counts of 5 or more and significant fold enrichment (FE) and false discovery rate (FDR) were included. The occurrence of each domain within the total human proteome (ALL) is listed as a reference. (**D**) Occurrence of intrinsically disordered proteins (IDPs; *Piovesan et al., 2023*) among SERBP1 interactors according to PFAM ('derived'=automatic entries from databases; 'curated'=manually checked). (**E**) Distribution of identified SERBP1-interacting proteins in reference to the presence of RGG boxes (*Thandapani et al., 2013*) and binding to G-quadruplexes (G4s) (*Su et al., 2021b*; *Herviou et al., 2020*). Data used to generate the figures are in *Supplementary file 1*, *Supplementary file 4* and *Supplementary file 6*.

The online version of this article includes the following source data and figure supplement(s) for figure 1:

**Figure supplement 1.** SERBP1-associated proteins isolated via pulldown experiments in 293T cells.

**Figure supplement 1—source data 1.** PDF file containing original protein gels for *Figure 1—figure supplement 1*.

**Figure supplement 1—source data 2.** Original protein gels for *Figure 1—figure supplement 1*.

**Figure supplement 2.** SERBP1 in translation and ribosome biogenesis.

*Figure 1 continued on next page*

*Figure 1 continued*

**Figure supplement 2—source data 1.** PDF file containing original western blots for *Figure 1—figure supplement 2A*.

**Figure supplement 2—source data 2.** Original western blots for *Figure 1—figure supplement 2A*.

**Figure supplement 3.** SERBP1-associated helicases and their respective domains.

proteins (SERBP1 homologs) interact with ribosomal proteins and proteins involved in RNA processing and transport (*Bleckmann et al., 2023*).

## SERBP1 preferentially associates with helicases and G-quadruplex binders

In order to define the characteristics of SERBP1-associated proteins, we cataloged them according to their protein domains using the Pfam database (*Sonnhammer et al., 1997*) and performed an enrichment analysis. RNA binding and helicase are the two domain families that appeared most often – *Figure 1C*, *Figure 1—figure supplement 3* and *Supplementary file 4*. Based on the results of the proteomics analyses and gene expression correlation studies, DDX21, DHX9, and DHX15 are SERBP1's top-associated helicases – *Supplementary file 5*.

SERBP1 displays large intrinsically disordered regions (IDRs; *Baudin et al., 2021*). IDRs are ubiquitous within biomolecular condensates in the cytoplasm and nucleus created via liquid-liquid phase separation (LLPS). They lead to the formation of membraneless organelles such as stress granules, P-bodies, and nucleoli (*Fonin et al., 2022*; *Su et al., 2021a*). Roughly 80% of SERBP1-associated proteins contain IDRs (*Piovesan et al., 2023*) and are present in different membraneless organelles, as discussed below – *Figure 1D* and *Supplementary file 4*.

SERBP1 is a G-quadruplex (G4) binder (*Kosti et al., 2020*; *Su et al., 2021b*), as are 38% of its associated proteins (*Su et al., 2021b*; *Herviou et al., 2020*) – *Supplementary file 6*. RBPs, including SERBP1, often use RGG boxes to bind G4 motifs (*Huang et al., 2018*; *Patra et al., 2021*). We found that among 93 SERBP1-associated proteins containing RGG boxes (*Thandapani et al., 2013*), 44 are G4 binders – *Figure 1E* and *Supplementary file 6*. G4 binders associated with SERBP1 are mostly proteins implicated in translation, mRNA stability, and splicing, indicating that SERBP1 regulation via G4 binding takes place in different scenarios. Among G4 binders, we identified several RNA helicases (e.g. DDX5, DDX21, DDX3X, and DHX9) that enhance translation by unfolding G4s in 5' UTRs (*Herdy et al., 2018*; *McRae et al., 2020*; *Murat et al., 2018*; *Varshney et al., 2021*) – *Figure 1—figure supplement 3*, *Supplementary file 1* and *Supplementary file 6*. Since SERBP1 lacks helicase domains, interaction with these proteins could be critical for SERBP1 function in translation.

G4s regulate different aspects of the life cycles of RNA viruses (*Ruggiero and Richter, 2018*; *Xu et al., 2021*). GO terms related to viral translation are among those with the highest fold enrichment in our analyses – *Supplementary file 1*. In a study to define the SARS-CoV-2 RNA interactome, 168 human proteins were identified (after exclusion of ribosomal proteins and EI3F factors; *Lee et al., 2021*), including SERBP1 and 92 of its associated proteins – *Supplementary file 6*. Of those 92 proteins, 52 are G4 binders – *Supplementary file 6*.

## SERBP1 affects cell division

SERBP1 has been previously implicated in chromosome segregation and shown to localize in M bodies during mitosis (*Martini et al., 2021*). In agreement, GO analyses of SERBP1-associated proteins revealed chromosome segregation as well as several related functions as enriched terms. Examination of proteins linked to these terms indicated that SERBP1 interacts with proteins in chromosomes as well as centrosomes and mitotic spindles – *Figure 2A* and *Supplementary file 1*. To evaluate SERBP1's impact on cell division, U251 and U343 SERBP1 knockdown and control cells were exposed to 20 nM of paclitaxel for 24 hr and later stained with α-tubulin and DAPI. In both cases, SERBP1 knockdown cells had markedly increased numbers of polynucleated cells, reflecting mitotic catastrophe – *Figure 2B–D*. SERBP1 involvement in cell division appears to be conserved. STM1, the SERBP1 yeast homolog, is a suppressor of mutations in genes regulating mitosis (*Chung et al., 2007*). Confirming that report, our GO analyses of STM1 genetic interactors showed enrichment for cell division – *Supplementary file 3*.

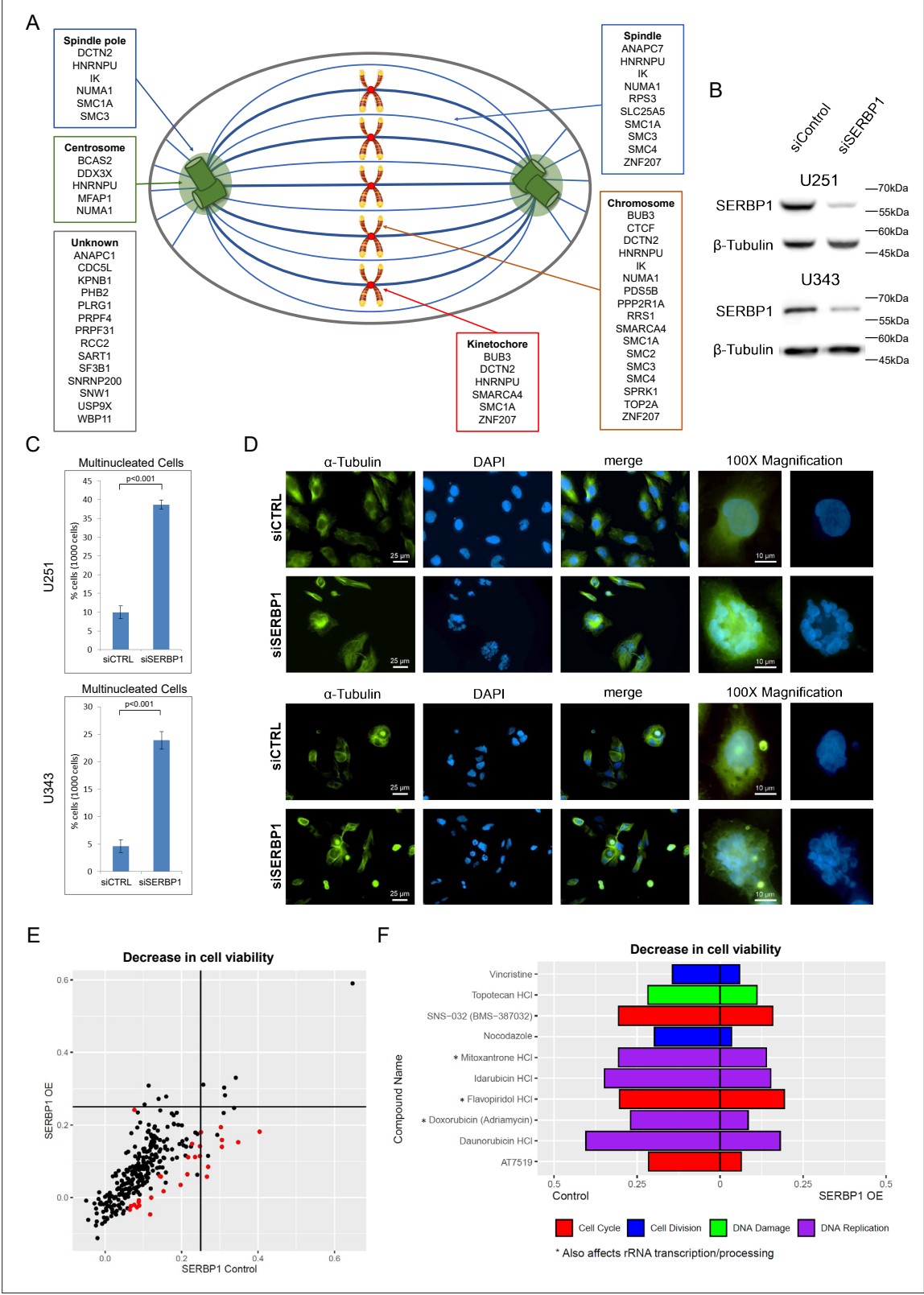

**Figure 2.** SERBP1 is implicated in cell division. (**A**) SERBP1-associated proteins and their presence in specific cellular structures relevant to mitosis. (**B**) Western blot showing SERBP1 knockdown in U251 and U343 cells. (**C**) Multinucleated cells in U251 and U343 control and SERBP1 knockdown (KD) cells after treatment with paclitaxel, an agent that causes cell cycle arrest. SERBP1 KD increased the number of multinucleated nuclei. Data are shown as means of counts of 1000 cells ± standard deviation (in triplicate) and statistical significance was determined by Student's t-test. *P*<0.001. (**D**) Aspect

*Figure 2 continued on next page*

*Figure 2 continued*

of cells exposed to paclitaxel. On the left staining with anti-α-Tubulin; in the middle staining with DAPI showing an increased number of multinucleated cells after siSERBP1 KD. ×100 magnification for detailed visualization of a single multinucleated cell. (**E**) Plot shows the results of a cell viability screening (291 drugs) performed in U343 control vs. U343 SERBP1 over-expressing (OE) cells. Red dots correspond to drugs whose impact on cell viability was significantly different between U343 control and U343 SERBP1 OE. (**F**) Highlights of the screening showing cell cycle/division and DNA damage/replication inhibitors whose impact on cell viability was smaller in U343 SERBP1 OE in comparison to control. Datasets used to prepare the figure and detailed analysis are in *Supplementary file 1* and *Supplementary file 7*.

The online version of this article includes the following source data for figure 2:

**Source data 1.** PDF file containing original western blots for *Figure 2B*.

**Source data 2.** Original western blots for *Figure 2B*.

We conducted a viability screening with 269 compounds in GBM U343 and U343 over-expressing (OE) SERBP1 (*Kosti et al., 2020*). Corroborating SERBP1 involvement in cell division, we determined that SERBP1 OE cells are less sensitive to drugs affecting cell division, cell cycle, and DNA replication. Interestingly, three of these drugs (Mitoxantrone, Flavopiridol, and Doxorubicin) are also known to affect rRNA transcription and processing – *Figure 2E and F* and *Supplementary file 7*.

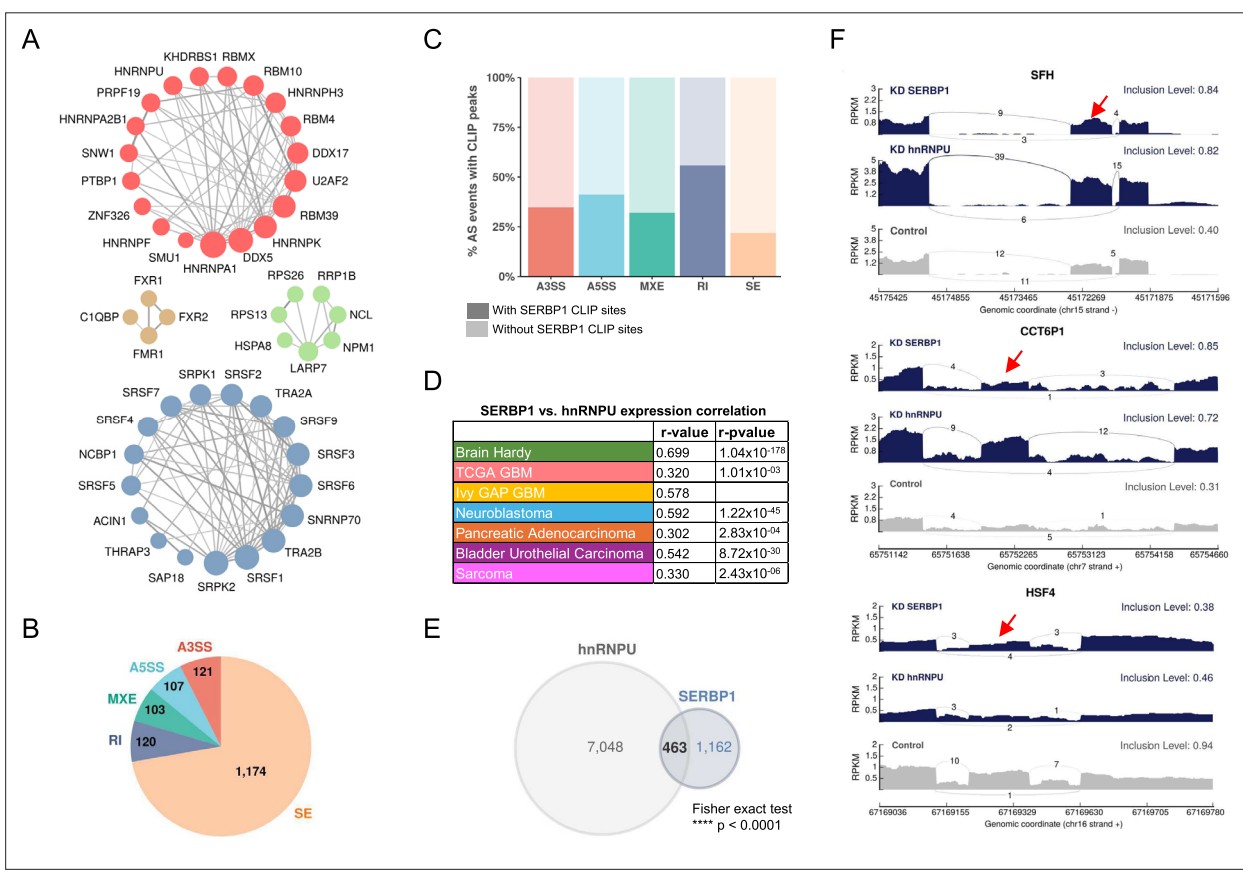

**Figure 3.** SERBP1 influences splicing. (**A**) SERBP1-interacting factors are preferentially associated with splicing. Networks show connections between splicing factors identified as SERBP1 interactors. (**B**) SERBP1 knockdown in U251 cells produced major changes in splicing. Pie chart displays the distribution of splicing events affected by SERBP1 knockdown according to their type: SE, exon skipping; RI, intron retention; MXE, multiple exclusive exons; A5SS, alternative 5' splice sites; A3SS, alternative 3' splice site. (**C**) Bar graphs showing the percentage of splicing events affected by SERBP1 knockdown with evidence of SERBP1 binding sites close (<100 nt) to regulated splice sites. (**D**) hnRNPU was identified as a potential partner of SERBP1 in splicing regulation. SERBP1 and hnRNPU display high expression correlation in normal and tumor tissues. (**E**) Venn diagram shows a strong overlap between the splicing events affected by SERBP1 and hnRNPU knockdowns in U251 cells. (**F**) Sashimi plots showing examples of splicing events affected by both SERBP1 and hnRNPU knockdowns in U251 cells. The red arrows indicate affected exons. Datasets used to prepare the figure, and detailed analyses are in *Supplementary file 8*.

## SERBP1 interacts with splicing factors and influences splice site selection

Among identified SERBP1 partners, 134 proteins (23%) are implicated in mRNA processing, and from this group, 117 participate in mRNA splicing. They include many well-known alternative splicing regulators such as hnRNPU, KHDRBS1, PTBP1, SRSF6, TRA2B, SRSF2, and hnRNPA1 – *Figure 3A*. In particular, hnRNPU, TRA2B, and KHDRBS1 display high expression correlation with SERBP1 in multiple scenarios – *Supplementary file 5*. To determine if SERBP1 expression levels influence splicing, we conducted an RNAseq study in U251 cells. SERBP1 knockdown altered the splicing of 1,625 events (PSI >0.1, FRD <0.05) with 70% of them being exon skipping events – *Figure 3B* and *Supplementary file 8*. Next, we examined a SERBP1 CLIPseq dataset (*Martini et al., 2021*) to search for evidence of SERBP1 direct participation in regulated splicing. From the 1625 splicing events altered in SERBP1 knockdown cells, CLIP sites were observed close to pertinent splice sites (<100 nucleotides) of 443 (27%) of them – *Figure 3C* and *Supplementary file 8*. SERBP1's association with many known alternative splicing regulators suggests that its impact on regulated splicing requires interaction with other factors. Since hnRNPU was identified as one of SERBP1's main partners and they display strong expression correlation in multiple instances (*Figure 3D*), we decided to conduct additional RNAseq analysis in U251 cells to determine if the two proteins co-regulate splicing events. Circa 30% of the events identified in SERBP1 knockdown cells were also observed upon hnRNPU knockdown, with changes in the same direction – *Figure 3E and F* and *Supplementary file 8*.

The results of the splicing analysis also revealed another potential route by which SERBP1 could affect ribosome biogenesis. Most expressed snoRNAs are embedded in introns of protein-coding genes and lncRNAs. Our analysis indicates that SERBP1 knockdown affects splicing events that include introns containing snoRNAs in 13 host genes – *Supplementary file 8*.

## SERBP1 has a two-way association with PARP1 and PARylation

We identified important associations between SERBP1 and PARP1, an enzyme that catalyzes the addition of poly(ADP-ribose) (PAR) on target proteins – *Figure 4A*. PARP1 is implicated in DNA repair and regulates multiple stages of gene expression, including transcription, splicing, ribosome biogenesis, and translation (*Kim et al., 2020*; *Gupte et al., 2017*; *Huang and Kraus, 2022*). SERBP1 and PARP1 showed strong expression correlated in several scenarios and similar expression profiles during cortex development – *Figure 4B and C*, *Supplementary file 5*.

Results of high throughput studies established that SERBP1 gets PARylated and binds to PAR (*Gibson et al., 2016*; *Martello et al., 2016*; *Dasovich et al., 2021*) – *Figure 4D*, *Supplementary file 6*. SERBP1 PARylation by PARP1 was confirmed in an in vitro assay – *Figure 4E*. Moreover, we determined that circa 56% of all identified SERBP1 interactors get PARylated and/or bind PAR (*Figure 4F*, *Supplementary file 6*), suggesting that PARylation is an important component in the assembly of protein complexes in which SERBP1 participates. We further investigated SERBP1-PARP1 interaction in 293T and U251 cells in the context of PARP activation. Cell exposure to $H_2O_2$ enhances PARP1 activity and protein PARylation as shown by Western blots probed with an anti-poly-ADP-ribose binding reagent – *Figure 4G*. In U251 cells, $H_2O_2$ treatment caused SERBP1 to shift to the nucleus, where it co-localized with PARP1 – *Figure 4—figure supplement 1*. Next, we co-transfected SBP-SERBP1-His and Flag-PARP1 in 293T cells and conducted pull-down experiments with streptavidin beads using extracts from cells treated with $H_2O_2$ and control. An increase in SERBP1-PARP1 interaction was observed in cells treated with $H_2O_2$ – *Figure 4H*. On the other hand, treatment with the PARP inhibitor PJ34 decreased SERBP1 interaction with PARP1 and G3BP1 as indicated by the results of the pulldown analysis with streptavidin beads – *Figure 4I*. Transgenic SERBP1 expression in 293T cells increased PARylation, as revealed by western blot probed with PAR detecting reagent – *Figure 4J*. To determine if PARylation/PAR binding affects SERBP1 interactions, we transfected 293T cells with an SBP-SERBP1 expressing vector. The experimental group was treated later with PJ34. SERBP1-associated proteins were isolated via pulldown with streptavidin beads and the presence of PARylated proteins was evaluated by Western blot. PARP inhibition decreased the amount of PARylated proteins associated with SERBP1 as shown in the pulldown lanes – *Figure 4—figure supplement 1*.

A recently published screening identified genes conferring sensitivity to PARP inhibitors, Olaparib, Rucaparib, and Talazoparib (*Zhang et al., 2023*). From this study, we generated a list with top hitters for each drug and compiled the results to identify genes appearing in two or more analyses. The 424

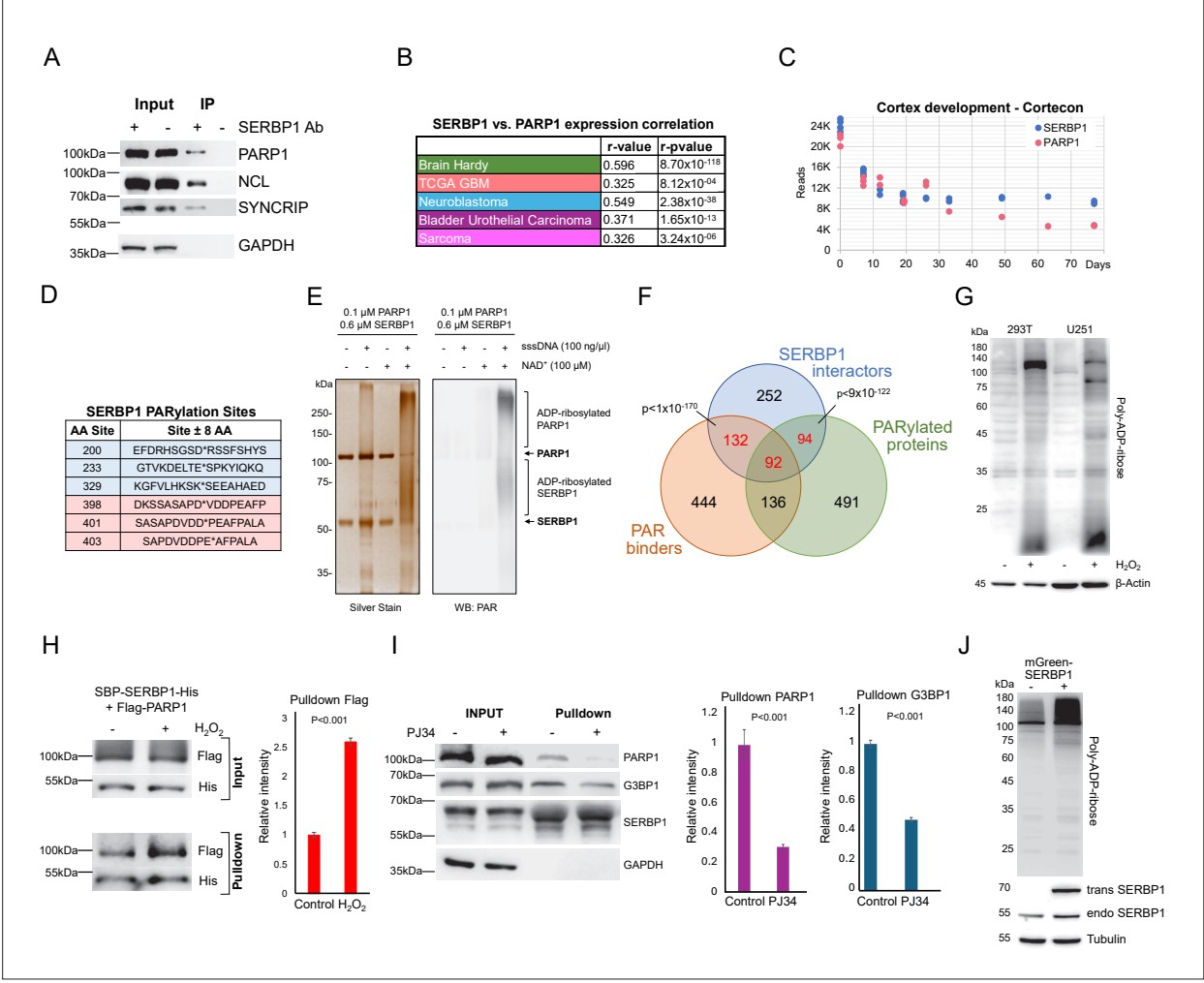

**Figure 4.** SERBP1 interacts with PARP1 and influences PARylation. (**A**) Results of IP-western in U251 cells with control and anti-SERBP1 antibodies confirm SERBP1 interaction with PARP1, NCL, and SYNCRIP. SERBP1 and PARP1 strong association is corroborated by their high expression correlation in different studies (**B**) and similar expression profiles during cortex development according to Cortecon (***van de Leemput et al., 2014***). (**C**). (**D**) PARylation sites observed in SERBP1 protein according to ***Gibson et al., 2016***; ***Martello et al., 2016***. (**E**) PARP1 ADP-ribosylates SERBP1 in vitro. Purified recombinant PARP1 (0.1 μM) and SERBP1 (0.6 μM) were combined in a reaction with or without sheared salmon sperm DNA (sssDNA) (100 ng/μL) and NAD+ (100 μM) as indicated. The reaction products were analyzed by SDS-PAGE with silver staining (left) and Western blotting for PAR (right). Uncropped gels are shown. (**F**) Venn diagram shows that majority of SERBP1-associated factors get PARylated and/or bind PAR (***Gibson et al., 2016***; ***Martello et al., 2016***; ***Dasovich et al., 2021***). (**G**) Increase of PARylation levels in 293T and U251 GBM cells after $H_2O_2$ treatment. (**H**) SBP-SERBP1 and Flag-PARP1 were co-transfected into 293T cells. Cells were treated with $H_2O_2$ to induce PARylation and a pull-down experiment with streptavidin beads was performed. Western analysis showed increased SERBP1 association with PARP1 in cells treated with $H_2O_2$. SBP-SERBP1-His detected by His antibody; Flag-PARP1 detected by Flag antibody. (**I**) SBP-SERBP1 was transfected into 293T cells. Cells were treated with DMSO or 10 μM PJ34 (PARP inhibitor) for two hours and a pull-down experiment with streptavidin beads was performed. Western analysis showed a decrease in SERBP1 association with PARP1 and GPBP1. (**J**) SERBP1 transgenic expression (mGreen-SERBP1) in 293T cells increased the levels of PARylated proteins as indicated by PAR-detecting agent. Datasets used to prepare the figure, and detailed analyses are in ***Supplementary file 1***, ***Supplementary file 5*** and ***Supplementary file 6***.

The online version of this article includes the following source data and figure supplement(s) for figure 4:

**Source data 1.** PDF file containing original western blots for ***Figure 4A***.

**Source data 2.** PDF file containing original western blots for ***Figure 4E***.

**Source data 3.** PDF file containing original western blots for ***Figure 4G***.

**Source data 4.** PDF file containing original western blots for ***Figure 4H***.

**Source data 5.** PDF file containing original western blots for ***Figure 4I***.

**Source data 6.** PDF file containing original western blots for ***Figure 4J***.

*Figure 4 continued on next page*

*Figure 4 continued*

**Source data 7.** Original western blots for *Figure 4*.

**Figure supplement 1.** PARP, PARylation and SERBP1 function.

**Figure supplement 1—source data 1.** PDF file containing western blots for *Figure 4—figure supplement 1B*.

**Figure supplement 1—source data 2.** Original western blots for *Figure 4—figure supplement 1B*.

identified genes include SERBP1 and 82 of its interactors – *Supplementary file 7*. In agreement, using a proliferation assay, we determined that SERBP1 partial knockdown makes U251 and U343 GBM cells more sensitive to the PARP inhibitor PJ34 – *Figure 4—figure supplement 1*.

PARP1 and several of its characterized interactors were determined to be associated with SERBP1 – *Supplementary file 1*. Our investigation has determined that the list of PARP1-SERBP1 shared factors is in fact more extensive. A comparison between SERBP1-associated factors and PARP1 interactors detected via proximity labeling (*Mosler et al., 2022*) identified 199 common factors (including SERBP1 and PARP1), of which about 80% are also PARylated and/or bind PAR (*Gibson et al., 2016*; *Martello et al., 2016*; *Dasovich et al., 2021*) – *Figure 5A* and *Supplementary file 6*. These proteins are implicated in splicing regulation, ribosome biogenesis, and DNA repair – *Figure 5B* and *Supplementary*

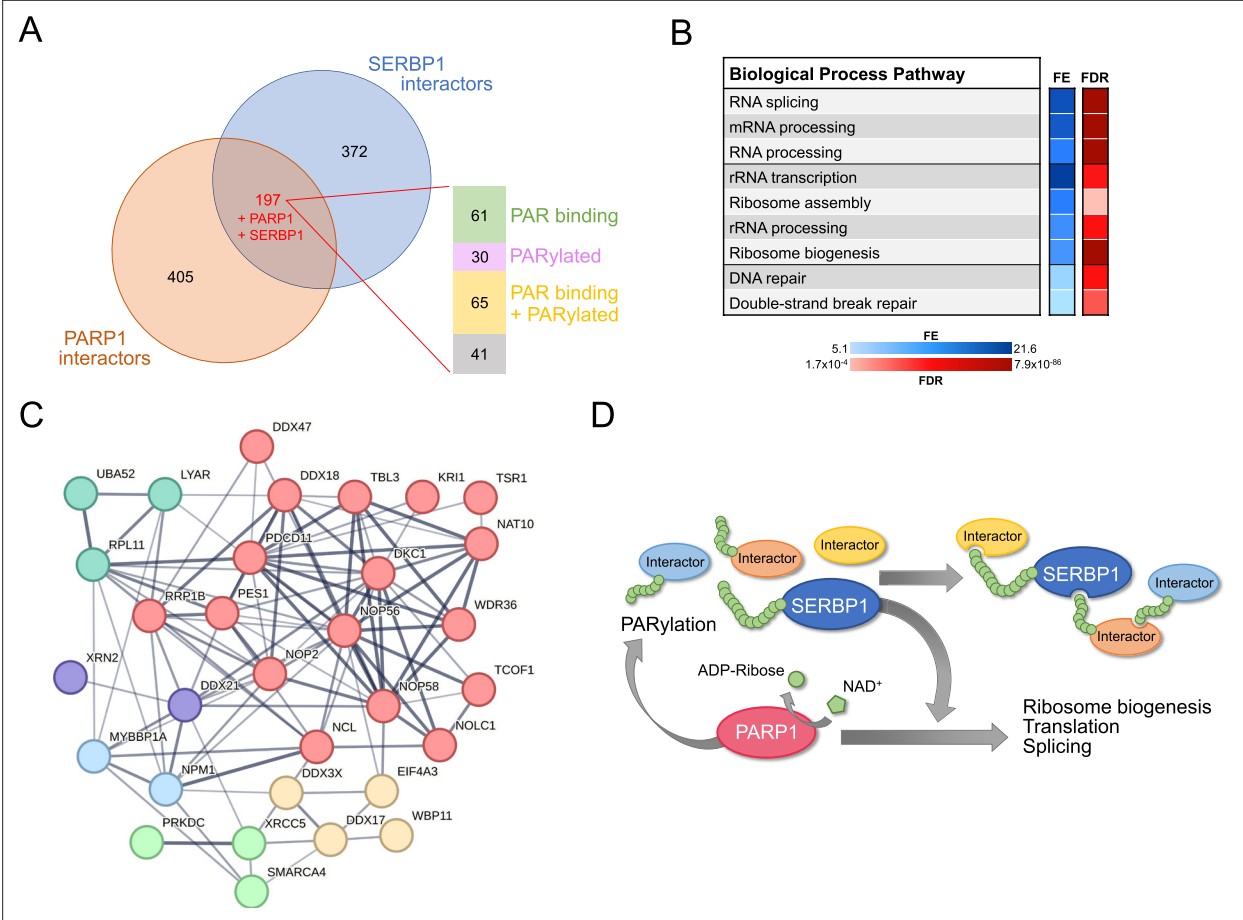

**Figure 5.** Shared SERBP1 and PARP1 interactors. (**A**) Venn diagram shows the overlap between PARP1 (*Mosler et al., 2022*) and SERBP1 interactomes. Analysis of shared interactors indicated that the majority of them are PAR binding (*Dasovich et al., 2021*) and/or get PARylated (*Gibson et al., 2016*; *Martello et al., 2016*). (**B**) Selection of enriched GO terms (biological processes) related to SERBP1-PARP1 shared interactors according to ShinyGO (*Ge et al., 2020*). FE = fold enrichment; FDR = false discovery rate. (**C**) Network showing shared SERBP1 and PARP1 interactors implicated in ribosome biogenesis. (**D**) Proposed SERBP1-PARP1 feedback model; SERBP1 function and association with partner proteins is modulated by PARylation while SERBP1 influences PARP activity. Datasets used to prepare the figure, and detailed analyses are in *Supplementary file 1* and *Supplementary file 6*.

The online version of this article includes the following figure supplement(s) for figure 5:

**Figure supplement 1.** SERBP1 interactors, PARylation, PAR- and G4-binding.

*file 6*. Among shared SERBP1-PARP1 interactors that get PARylated and/or bind to PAR are multiple key factors implicated in rRNA transcription, processing, and modification, including NOLC1, NOP16, NAT10, DKC1, DDX21, NCL, SMARCA4 and TCOF1 – *Figure 5C* and *Supplementary file 6*. The binding of snoRNAs to PARP1 stimulates its catalytic activity in the nucleolus independent of DNA damage. Activated PARP1 PARylates the RNA helicase DDX21 to promote rDNA transcription (*Kim et al., 2019*). DDX21 was identified as a top SERBP1-associated factor, and expression of the two proteins was highly correlated in several scenarios – *Supplementary file 1* and *Supplementary file 5*. Moreover, we and others have determined that SERBP1 preferentially binds snoRNAs, including the ones implicated in PARP1 activation: snoRA37, snoRA74A, and snoRA18 (*Kim et al., 2019*). Out of 58 snoRNAs bound by PARP1 (*Mekishvili et al., 2017*), 32 of them are also targeted by SERBP1 – *Supplementary file 2*.

Adding up to SERBP1-PARP1 strong functional association, we determined that about 50% of SERBP1 interactors known to bind PAR, are also G4 binders – *Figure 5—figure supplement 1A*, *Supplementary file 6*. Different types of protein domains bind to PAR, including RGG boxes and RRMs (*Wei and Yu, 2016*; *Zaja et al., 2012*). These two types of domains are also among the ones that bind G4s. About half of SERBP1-associated proteins that were identified as both PAR and G4 binders contain RGG boxes and/or RRMs (*Thandapani et al., 2013*; *Bateman et al., 2023*) – *Figure 5—figure supplement 1B* and *Supplementary file 6*. Based on our findings, we propose the existence of a feedback regulatory model in which SERBP1 influences PARP1 function and PARylation, while PARylation modulates SERBP1 functions and association with its protein partners – *Figure 5D*.

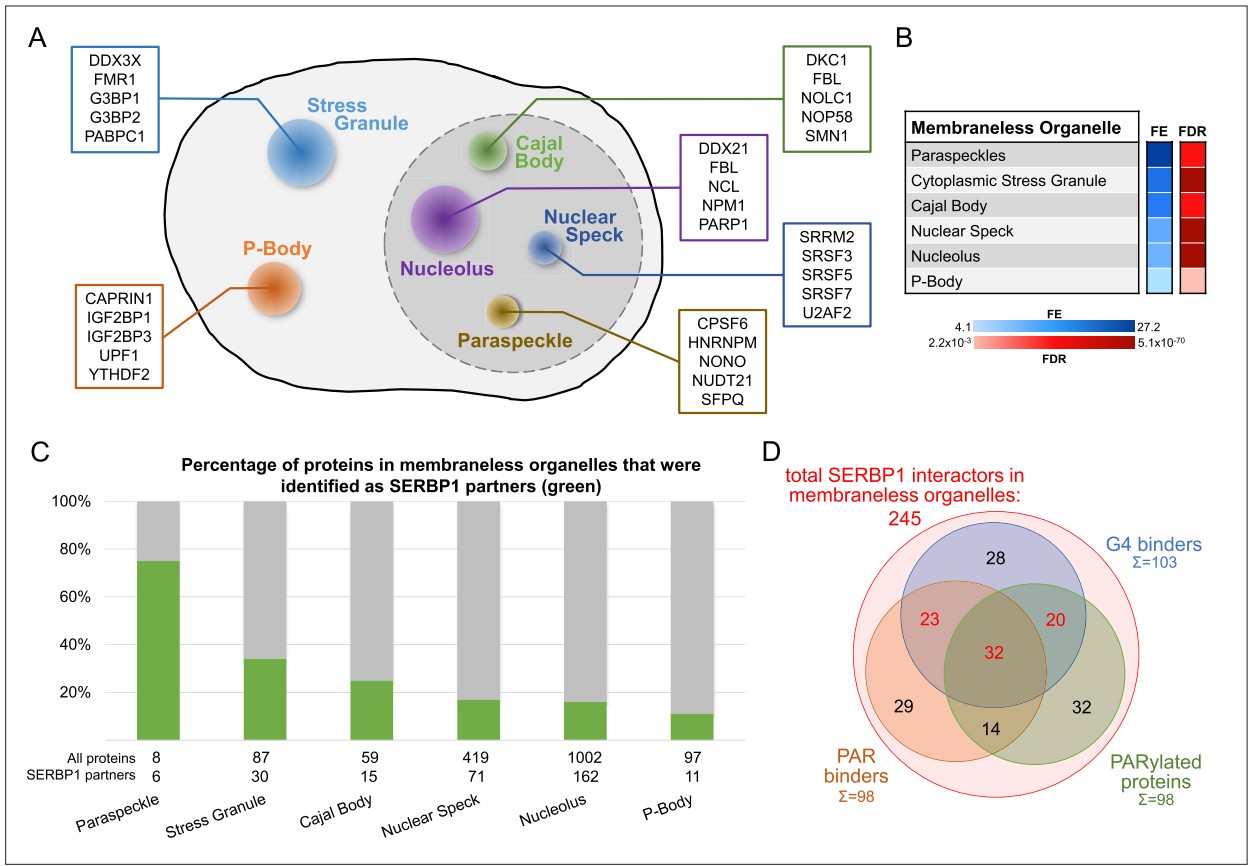

**Figure 6.** SERBP1-associated proteins are present in membraneless organelles. (**A**) Different types of membraneless organelles in a cell and examples of SERBP1-associated proteins present in each structure. (**B**) Membraneless organelles identified in the GO enrichment analysis (cellular component) of SERBP1 interactors (*Thomas et al., 2022*). FE = fold enrichment; FDR = false discovery rate. (**C**) Bar graph showing the distribution of SERBP1-associated proteins in different membraneless organelles. (**D**) Venn diagram representing the distribution of SERBP1-associated proteins present in membraneless organelles in respect to G4 binding (*Su et al., 2021b*; *Herviou et al., 2020*), PAR binding (*Dasovich et al., 2021*), and PARylation (*Gibson et al., 2016*; *Martello et al., 2016*). Datasets used to prepare the figure, and detailed analyses are in *Supplementary files 1, 6 and 9*.

## SERBP1 is in association with proteins present in membraneless organelles and Tau aggregates

GO analysis of SERBP1-associated proteins based on cellular compartment (*Thomas et al., 2022*) determined that SERBP1 interacts with factors often present in membraneless organelles, including stress granules, P bodies, Cajal bodies, nuclear speckles, paraspeckles, and nucleoli, which suggests SERBP1's likely participation in protein complexes present in these organelles – *Supplementary file 9* and *Figure 6*. SERBP1 binding to snoRNAs and scaRNAs further supports its presence in Cajal bodies and nucleolus – *Supplementary file 2*. Liquid-liquid phase separation (LLPS) is a main contributor to the formation of membraneless organelles or granules (*Li et al., 2021*). We have previously established that SERBP1 forms biomolecular condensates in vitro that are modulated by RNA (*Baudin et al., 2021*). A characteristic of proteins in LLPS is the presence of IDRs (*Kastano et al., 2022*; *Feng et al., 2021*). Of all SERBP1-associated proteins present in membraneless organelles, 77% of them contain IDRs – *Supplementary file 9*.

SERBP1 has been observed in Tau aggregates in Alzheimer's disease (AD) brains and is part of a group of 261 proteins detected in several studies that examined the composition of Tau aggregates (*Kavanagh et al., 2022*). We further investigated SERBP1's association with pathological Tau by co-staining SERBP1 in AD tissues with phospho-tau (Thr231), which is associated with large inclusion and paired helical filaments (PHF) (*Kimura et al., 2018*). We found a signal overlap between large hyperphosphorylated Tau inclusions and SERBP1 puncta (*Figure 7A*), confirming their close interaction in AD tissue. The interaction between SERBP1 and pTau was confirmed using an in situ proximity ligation assay – *Figure 7B*. Quantification of the PLA signal showed increased SERBP1-pTau interaction in AD compared to Ctr – *Figure 7C*. Western blot analysis showed that SERBP1 displays increased expression in AD brains in comparison to normal controls. Interestingly, SERBP1 appears to be often present as a dimer – *Figure 7D and E*. Overall, our data suggest that SERBP1 shows increased accumulation in AD brains and condenses in hyperphosphorylated Tau aggregates.

146 proteins present in Tau aggregates were also identified as SERBP1 interactors – *Figure 7F* and *Supplementary file 9*. GO analysis indicated that this group is strongly associated with ribosome biogenesis, translation, and splicing – *Figure 7F*. snoRNAs are also aberrantly located in Tau aggregates (*Lester et al., 2021*). Adding to SERBP1 participation in Tau aggregates, we determined that 66% of the snoRNAs enriched in Tau aggregates were bound by SERBP1 according to our study – *Supplementary file 9*. Ribosome biogenesis and translation are impaired in AD (*Hernández-Ortega et al., 2016*; *Evans et al., 2021*). SERBP1 may help trap snoRNAs and factors implicated in ribosome biogenesis and translation in Tau aggregates, but further investigation is required.

PARP1 activity is increased in AD brains. Immunostaining of PARP1 and PAR indicates that PARylation is hyperactivated in AD neurons. Ultimately, this hyperactivation can trigger proteins to condense into toxic aggregates that contribute to neurodegeneration (*Mao and Zhang, 2022*). We investigated PARP1-SERBP1 association in normal vs. AD brains. PARP1 exhibited a stronger presence in AD cortices compared to controls. Furthermore, PARP1 appeared to co-localize with SERBP1 in the cytoplasmic fraction of AD samples (white arrowhead) – *Figure 8A*. In control samples, PARP1 and SERBP1 also co-localized, albeit to a lesser extent compared to AD samples (confirmed by PCC analysis) – *Figure 8A and B*. Fluorescence profiles corroborate their association – *Figure 8C*. We performed a PLA assay to measure physical interaction between SERBP1 and PARP1. PLA results showed an increased signal for SERBP1/PARP1 association in AD brains in comparison to control brains. This observation was confirmed by the quantification of positive PLA area covered by such association – *Figure 8D*.

G3BP1, a well-established regulator of stress granule assembly, has been identified here as a SERBP1-associated protein – *Supplementary file 1*. In AD brains, G3BP1 is present in pathological stress granules (*Ash et al., 2014*; *Vanderweyde et al., 2012*). Co-immunofluorescence (IF) experiments in control and AD brain sections revealed a stronger fluorescent signal for SERBP1 and G3BP1 in AD compared to controls, indicating higher levels of SERBP1 protein in stress granules in pathological condition – *Figure 8—figure supplement 1A*. Moreover, SERBP1 IF showed a high density of puncta with some large aggregates in AD samples. To quantify the level of SERBP1 in stress granules in control and AD samples, we measured the number of positive SERBP1 puncta in G3BP1 stress granules, observing a significant increment of SERBP1 in stress granules in AD cases. The total number of G3BP1 and SERBP1 puncta indicated greater accumulation in AD tissues compared to control

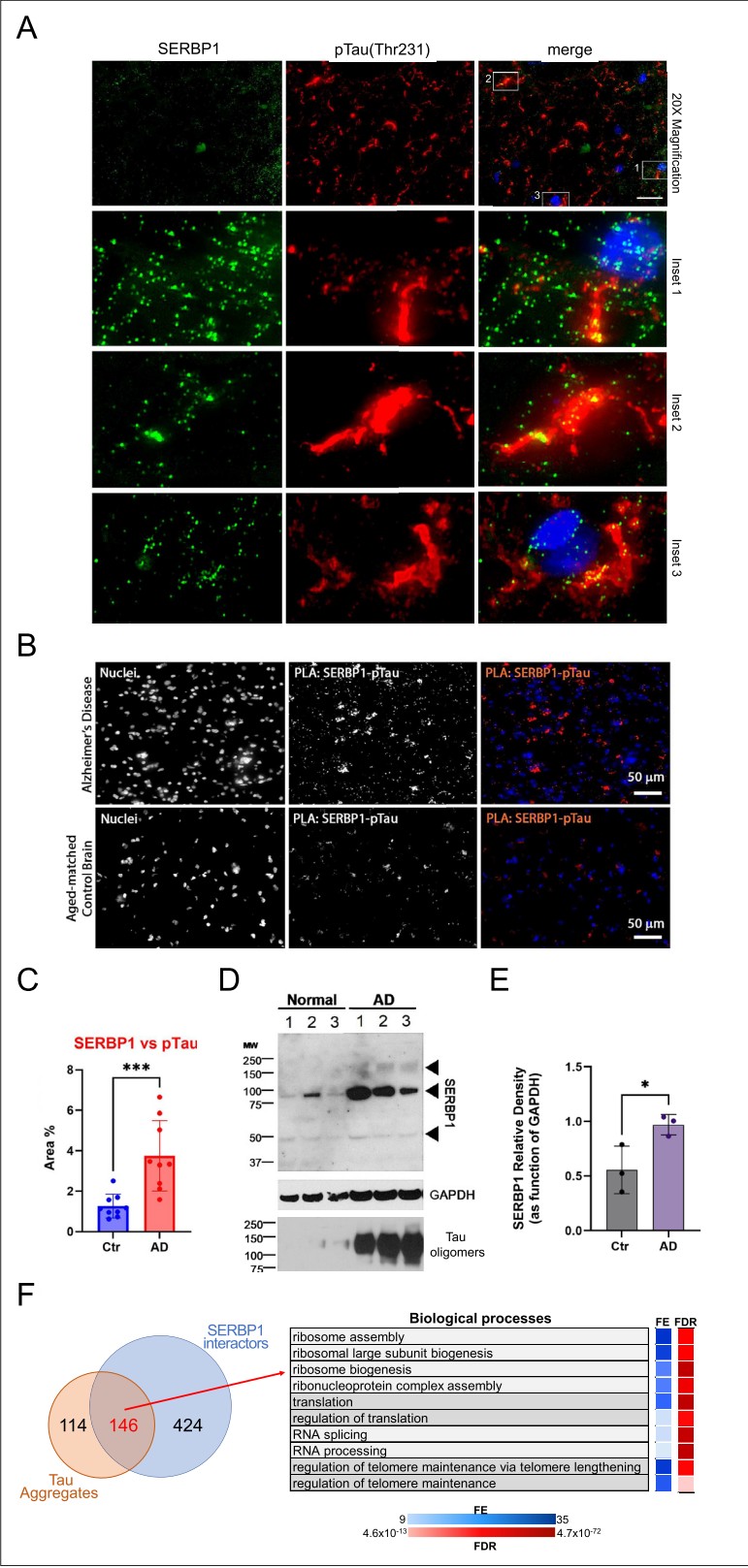

**Figure 7.** SERBP1 is present in Tau aggregates. (**A**) Representative co-immunofluorescence of SERBP1 and Phospho-Tau (Thr231) in AD brain tissues. Merged channel is represented. DAPI was used to stain nuclei. Magnification 20 x and white scale bar: 50 μm. Three different Insets selected from merged channels are represented in zoomed images as merged, SERBP1 (green) and Phospho-Tau (red). (**B**) Representative PLA

*Figure 7 continued on next page*

*Figure 7 continued*

of SERBP1 vs. pTau in AD and aged-matched control brains (magnification 40 x and white scale bar: 50 µm). (**C**) Percentage of positive area to PLA fluorescence in region of interests in AD and control brains (Ctr vs AD, *** p<0.001, paired t-test). (**D**) Western blot showing SERBP1 expression in normal and AD brains and presence of oligomers. GAPDH and Tau oligomers immunoblot are shown. (**E**) SERBP1 relative density in Ctr and AD brains, quantified as function of GAPDH (from immunoblot in D). (Ctr. vs. AD, * p<0.05, paired t-test). (**F**) Venn diagram shows overlap between proteins often identified in Tau aggregates and SERBP1 associated proteins identified in this study (top). Gene Ontology (GO) enrichment analysis (Biological Processes) using ShinyGO (*Ge et al., 2020*) indicated that SERBP1-associated factors that are also present in Tau aggregates are implicated in ribosome biogenesis, translation, splicing and telome maintenance. FE = fold enrichment; FDR = false discovery rate. Datasets used to prepare the figure, and detailed analyses are in *Supplementary file 1* and *Supplementary file 9*.

The online version of this article includes the following source data for figure 7:

**Source data 1.** PDF file containing original western blots for *Figure 7D*.

**Source data 2.** Original western blots for *Figure 7D*.

– *Figure 8—figure supplement 1B*. SERBP1 and G3BP1 also show strong co-localization in glioblastoma cells and highly correlated expression in different scenarios – *Figure 8—figure supplement 1C*, *Supplementary file 5*.

## Discussion

SERBP1's name relates to its function as a regulator of the gene SERPINE1, which encodes the plasminogen activator inhibitor 1 (PAI-1) (*Heaton et al., 2001*), a protein implicated in senescence and migration. SERBP1 is localized mostly in the cytoplasm, but it is also present in the nucleus. SERBP1 cytoplasmic/nuclear balance changes during the cell cycle, and its nuclear presence intensifies upon stress (*Ahn et al., 2015*; *Lee et al., 2014*; *Passos et al., 2006*). We have identified SERBP1's interactome and expanded its regulatory roles – particularly in the nucleus, where it is associated with specific complexes implicated in splicing, cell division, chromosome structure, and different aspects of ribosome biogenesis.

Most SERBP1-associated proteins are also RBPs. However, SERBP1 has unique characteristics; it is mostly disordered and does not display classic RNA binding domains other than RGG boxes (*Baudin et al., 2021*). Two commonalities between SERBP1 and its associated proteins are their preference for G4 binding and the fact that they get PARylated and/or bind to PAR. These results indicate that complexes in which SERBP1 participates are assembled via G4 or PAR binding. Since SERBP1 does not display any apparent functional domains and is involved in very distinct regulatory complexes, we suggest that SERBP1 acts in the initial steps of their assembly via recognition of interacting sites in RNA, DNA, and proteins.

### SERBP1 in translation regulation and ribosomal biogenesis

SERBP1 interacts with ribosomal proteins and regulates translation (*Brown et al., 2018*; *Muto et al., 2018*; *Ahn et al., 2015*; *Martini et al., 2021*), functioning as a repressor or activator depending on the context. For instance, SERBP1 binding to CtIP mRNA during the S phase increases its translation (*Ahn et al., 2015*), while changes in binding to the 40 S ribosomal subunit during mitosis lead to PKCε pathway-mediated translation repression (*Martini et al., 2021*). Additionally, the SERBP1 yeast homolog, STM1, is required for optimal translation under nutritional stress (*Van Dyke et al., 2006*). SERBP1 migrates to stress granules under stress conditions (*Lee et al., 2014*). Recent findings indicated that translation in these organelles is not uncommon (*Mateju et al., 2020*), suggesting the possibility of a similar role for SERBP1 as a modulator of translation in stress conditions.

Our GO analyses revealed that SERBP1 interacts with proteins participating in all stages of translation – initiation, elongation, and termination – including both negative and positive regulators. We found that 17% of SERBP1-associated proteins are implicated in translation initiation, including several subunits of the eukaryotic initiation factor 3 (eIF3A-I, L, and M) and CSDE1, among others. These data suggest that SERBP1's primary function in translation regulation lies in initiation. CSDE1 is a known regulator of both cap-dependent and cap-independent translation, functioning either as an

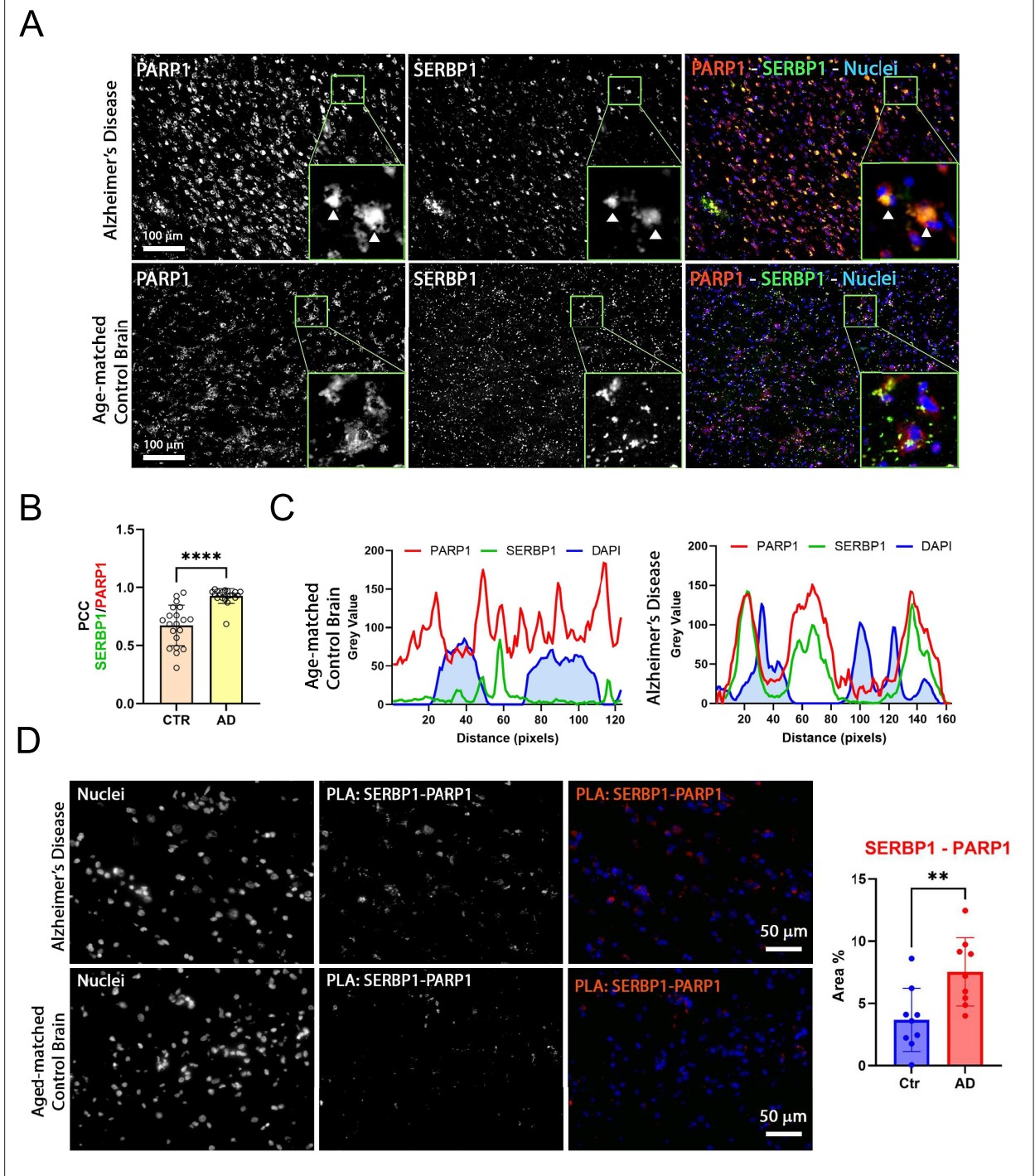

**Figure 8.** SERBP1 association with PARP1 in Alzheimer's brains. (**A**) Representative co-immunofluorescence of SERBP1 and PARP1 in AD and age-matched control brain tissues. PARP1 and SERBP1 are represented in gray while merged images represent PARP1 (red), SERBP1 (green). DAPI was used to stain nuclei (blue). Magnification ×20 and white scale bar: 100 μm. Each frame has a zoomed inset representing the detailed distribution of each target. (**B**) Pearson Coefficient (PCC) of co-localizing SERBP1 and PARP1 between cells of age-matched control and AD brains (CTR vs. AD, **** p<0.001 paired t-test). (**C**) Fluorescence intensity profiles of PARP1 (red), SERBP1 (green), and DAPI (blue) in representative cells from age-matched control and AD brains. Distance is represented in pixels and fluorescence intensity as Grey value obtained using ImageJ FIJI software. (**D**) Representative PLA of SERBP1 vs. PARP1 in AD and aged-matched control brains (magnification 40 x and white scale bar: 50 μm). Percentage of positive area to PLA fluorescence in region of interests in AD and control brains (Ctr vs AD, ** p<*0.01*, paired t-test).

The online version of this article includes the following figure supplement(s) for figure 8:

**Figure supplement 1.** SERBP1 association with G3BP1 in pathological stress granules and glioblastoma cells.

enhancer (*Ray and Anderson, 2016*; *Mitchell et al., 2001*) or repressor (*Dormoy-Raclet et al., 2005*; *Abaza et al., 2006*). The expression of CSDE1 is highly correlated with SERBP1 in multiple instances. CSDE1 was proposed to function as a 'protein-RNA connector' enabling interactions between RNAs and regulators to control RNA's translational fate (*Guo et al., 2020*). A similar role could be ascribed to SERBP1, possibly explaining its different effects on translation.

Multiple SERBP1-associated proteins are involved in viral translation. For instance, the aforementioned interactors, components of the eIF3 complex and CSDE1, are both implicated in IRES-mediated viral translation in a variety of viruses (*Walsh and Mohr, 2011*; *Locker et al., 2011*; *Boussadia et al., 2003*). SERBP1 itself has also recently been shown to interact with dengue virus RNA and regulate its translation (*Brugier et al., 2022*). Finally, we observed a significant overlap between SERBP1-associated proteins and the SARS-CoV-2 interactome, with 82 proteins out of 145 shared interactors being involved in translation.

SERBP1's interactors and its presence in the nucleolus suggest its participation in ribosome biogenesis. Besides binding to mRNA, SERBP1 binds to rRNA and snoRNAs, which play a role in pre-rRNA processing and modification (*Tollervey and Kiss, 1997*). SERBP1, and 98 of its associated proteins, were previously identified in a screening for factors affecting rRNA processing (*Tafforeau et al., 2013*). SERBP1's binding profile to rRNA and snoRNA is dynamic, as shown by differences in iCLIP sites in mitotic vs. asynchronous cells (*Martini et al., 2021*), hinting that SERBP1's participation in these processes is cell cycle-dependent as seen for translation. FBL, a SERBP1-associated protein, is involved in pre-rRNA processing, pre-rRNA methylation, and ribosome assembly (*Tollervey et al., 1993*). Recently, it has been shown that FBL also modulates rDNA transcription during the cell cycle according to its acetylation status, leading to higher levels of rRNA synthesis in interphase and lower levels during mitosis (*Iyer-Bierhoff et al., 2018*). SERBP1 also interacts with multiple proteins of the cohesin complex (SMC1, SMC3, and PDS5B) plus the associated factor, CTCF. Cohesin forms an enhancer boundary complex who's binding upstream of the 47 S rDNA is as an early event in the activation of rDNA transcription (*Herdman et al., 2017*). In accordance with this finding, mutations in cohesin components have been linked to defects in both rRNA production and processing in yeast and human cells (*Bose et al., 2012*). Two of SERBP1's top interactors, the RNA helicase DDX21 and NCL, also promote rDNA transcription (*Calo et al., 2015*; *Ugrinova et al., 2018*).

Overall, these findings indicate a rather fluid role for SERBP1 in translation regulation, promoting or repressing different translational steps and acting in rRNA processing/modification and ribosome biogenesis through diverse pathways.

## SERBP1 and PARylation

PARylation of SERBP1 was described in several different studies (*Gibson et al., 2016*; *Martello et al., 2016*; *Hendriks et al., 2019*; *Bonfiglio et al., 2017*; *Bilan et al., 2017*; *Larsen et al., 2018*). Moreover, the expression of human PARP1 in *S. cerevisae,* which does not naturally express PARP proteins, caused PARylation of many proteins implicated in ribosomal biogenesis, including STM1, SERBP1 homolog (*Tao et al., 2009*). Our results suggest a two-way association between SERBP1 and PARP1. First, SERBP1 and 56% of its identified associated proteins either get PARylated and/or bind PAR (*Gibson et al., 2016*; *Martello et al., 2016*; *Dasovich et al., 2021*). This result suggests that SERBP1 participates in several regulatory complexes modulated via PARylation.

PARylation of RBPs ultimately influences RNA processing and expression (*Kim et al., 2020*; *Manco et al., 2022*). In particular, PARylation of splicing regulators is known to affect alternative splicing. For instance, PARylation of HRP38's RNA binding domain decreases its binding activity and subsequently its ability to inhibit splicing (*Kim et al., 2020*). PARylation also affects protein-protein interaction as observed in the case of hnRNPA1 (*Duan et al., 2019*). Finally, PARP1 itself is involved in splicing. PARP1 binds RNA with a preference for intronic regions, while its knockdown affected many splicing events (*Melikishvili et al., 2017*). Several splicing regulators known to get PARylated and/or bind to PAR were identified as SERBP1-associated proteins, including hnRNPs (A1, H1, A2B1, H3, K, I, and U) and SR proteins (SRSF2, SRSF3, and SRSF9). In particular, most splicing alterations observed in SERBP1 KD cells were also present upon hnRNPU KD, suggesting that they work as co-factors in splicing regulation. Altogether, the data suggest that SERBP1 is directly implicated in splicing regulation and is present in complexes modulated by PARylation.

PAR induces phase separation of proteins containing IDRs and the formation of molecular condensates or membraneless organelles such as stress granules, mitotic spindles, DNA repair foci, and nucleoli (*Leung, 2020*). Similarly, ADP-ribose can be conjugated to polar and charged amino acids, which are often found in IDRs (*Leung, 2020*). Ultimately, PAR serves as a scaffold to assemble protein complexes in different contexts where PAR length and structure influence their composition (*Leung, 2020*). The large number of PARylated proteins and PAR binders found among SERBP-associated proteins strongly indicates that PAR is a critical component of the regulatory complex in which SERBP1 participates, especially in membraneless organelles. As a protein that interacts with PARP1, promotes PARP activity, gets PARylated, and binds PAR, we suggest that SERBP1 acts as a critical factor in the assembly of PARylation-dependent complexes. Additional analyses are required to assess whether the formation of such complexes depends on SERBP1 and if inhibition of SERBP1 PAR binding activity could be explored as a therapeutic strategy.

SERBP1 exhibited a punctate distribution in normal brain, indicating its presence as functional singlet condensate. However, in AD brain tissues, SERBP1 appeared to accumulate in large cytoplasmic aggregates and to strongly co-localize with PARP1. This result suggests that PARP1 modulates liquid-liquid phase separation (LLPS) and a PARylation-mediated mechanism is likely involved in SERBP1's pathological transition from condensate to aggregate. Several RBPs containing intrinsically disordered regions, like SERBP1, have been shown to display alterations in LLPS in neurodegenerative diseases (*Ottoz and Berchowitz, 2020*; *Naskar et al., 2023*).

## SERBP1 and G4 binders

SERBP1 belongs to a group of neuronal-related RBPs that share structural characteristics and preferentially bind to G quadruplexes (G4s). G4s are complex DNA or RNA structures that display stacked tetrads of guanosines stabilized by Hoogsteen base pairing. rG4s function as regulatory elements in different stages of gene expression, including splicing, mRNA decay, polyadenylation, and translation. G4s have been linked to cancer, viral replication, and in particular neurodegenerative diseases and neurological disorders (*Lyu et al., 2021*; *Banco and Ferré-D'Amaré, 2021*; *Dumas et al., 2021*).

The SERBP1 interactome shows an interesting parallel between PAR and G4 binding. Almost half of the SERBP1-associated proteins that bind G4 also bind PAR. PAR and G4s occupy the same regulatory space that includes DNA repair and stress response, and both induce phase separation. However, not much is known regarding the association between G-quadruplex and PAR binding/PARylation, aside from the fact that PARP1 binds G4s, and this leads to the activation of its enzymatic activity (*Edwards et al., 2021*). We envision two possible scenarios for SERBP1 and its associated proteins. Two types of domains have been implicated in both PAR and G4 binding, RGG boxes and RRMs. Among 105 SERBP1-associated factors that bind PAR and G4, 51 contain RGG and/or RRM domains. Proteins that employ the same domain to bind PAR and G4 could transit from G4 to PAR binding complexes, depending on the cellular state. In a different setting, binding to G4 motifs in DNA or RNA could serve as a starting point for the assembly of PAR binding/PARylated protein complexes. Further investigation on PAR- and G4-dependent interactions and identification of G4 and PAR binding domains in proteins present in these complexes are needed to elucidate the relationship between PAR and G4 binding.

## SERBP1 potential role in neurodegenerative diseases and neurological disorders

SERBP1 expression in the brain is lower than in other organs and is increased in glioblastoma (*Kosti et al., 2020*) and Alzheimer's brains, as shown in this study. SERBP1 is present in Tau aggregates (*Kavanagh et al., 2022*) and in pathological stress granules in connection with G3BP1. PAR levels regulate the dynamics of RNPs and aberrant PARP1 activity promotes aggregation (*Duan et al., 2019*). Over 60% of SERBP1-associated proteins detected in Tau aggregates get PARylated and/or bind PAR (*Gibson et al., 2016*; *Martello et al., 2016*; *Dasovich et al., 2021*) and increased SERBP1-PARP1 interaction was observed in AD brains. Besides its potential role in aggregate formation, increased expression of SERBP1 in Alzheimer's brains could lead to important changes in gene expression. SERBP1 is highly expressed in neuronal and glioma stem cells and its levels drop significantly during neurogenesis. SERBP1 transgenic expression disrupted neuronal differentiation while its knockdown

increased the expression of genes implicated in nervous system development, synaptic signaling, organization, memory, learning, and behavior (*Kosti et al., 2020*).

SERBP1-associated proteins contain several factors involved in neurological disorders and neurodegenerative diseases, including FMR1, SMN1, and PABPN1. SERBP1 interaction with FMRP (FMR1) has been previously described (*Martini et al., 2021*; *Lee et al., 2014*). Not only was this interaction validated, but we also determined that SERBP1 associates with FMRP partner proteins FXR1 and FXR2. FMRP is the main driver of Fragile X mental retardation but has also been linked to Alzheimer's disease and autism (*Bleuzé et al., 2021*; *Fyke and Velinov, 2021*). FMRP is a complex protein that binds different RNA motifs (including G4s) and regulates translation, splicing, RNA editing, and miRNA function, affecting synapse formation and plasticity, stress granule formation, channel signaling, and differentiation (*Davis and Broadie, 2017*; *Darnell et al., 2004*; *Darnell and Klann, 2013*; *Richter and Zhao, 2021*). Previous results indicate that SERBP1 and FMRP interact in different contexts (*Martini et al., 2021*; *Lee et al., 2014*), suggesting that they function together in diverse stages of gene expression.

Spinal Muscular Atrophy (SMA) is caused by mutations in the SMN1 gene. Patients show progressive lower motor neuron loss (*Prashad and Gopal, 2021*). We observed associations between SERBP1, SMN1, and other members of the SMN complex, GEMIN4 and 5. SMN1 brings together the complex and is essential for spliceosome assembly. Since motor neurons are particularly sensitive to SMN1 mutations, additional roles for the SMN complex in these cells are proposed (*Prashad and Gopal, 2021*).

SYNCRIP and hnRNPU were identified among the top proteins associated with SERBP1 based on our proteomics analyses and gene expression correlations. SYNCRIP plays multiple roles in neuronal cells, including dendritic translation, synaptic plasticity, axonogenesis, and morphology (*Chen et al., 2012*; *Duning et al., 2008*; *Khudayberdiev et al., 2021*; *Titlow et al., 2020*; *Halstead et al., 2014*). hnRNPU has been implicated in cortex development (*Sapir et al., 2022*). Mutations in both hnRNPU and SYNCRIP are linked to intellectual disability (*Lelieveld et al., 2016*; *Balasubramanian, 2022*; *Bramswig et al., 2017*) and neurodevelopmental disorders (*Gillentine et al., 2021*; *Wang et al., 2020*), and were the two top candidates in an IBM Watson-based study to identify RBPs altered in amyotrophic lateral sclerosis (ALS) (*Bakkar et al., 2018*).

SERBP1 has not been studied in the context of the nervous system, and its contribution to neurodegenerative diseases and neurological disorders remains to be investigated. Our results suggest that SERBP1 is 'guilty by association' as it interacts with several RBPs that function as key players in these scenarios. Moreover, SERBP1 regulates the expression of genes implicated in neuronal differentiation and synaptogenesis, processes often compromised in disease states. In the case of Alzheimer's disease, increased expression of SERBP1 and its presence in pathological aggregates hint at a potential involvement. Additional studies are necessary to establish causal connections that link SERBP1 to AD phenotypes.

## Materials and methods

### Key resources table

| Reagent type (species) or resource | Designation | Source or reference | Identifiers | Additional information |
|---|---|---|---|---|
| Gene (*Homo sapiens*) | SERBP1 | GenBank | HGNC:HGNC:17860 | |
| Cell line (*H. sapiens*) | U251 (glioblastoma) | Uppsala University | | |
| Cell line (*H. sapiens*) | U343 (glioblastoma) | Uppsala University | | |
| Cell line (*H. sapiens*) | 293T (normal) | ATCC | CRL-3216 | |
| Transfected construct (human) | SERBP1 SMARTpool siRNA | Dharmacon | Cat#:L-020528-01-0005 | |
| Biological sample (human) | Control and Alzheimer's Disease (AD) human brains | Institute for Brain Aging and Dementia at UC Irvine | | Braak Stage V-VI for AD brains |

*Continued on next page*

*Continued*

| Reagent type (species) or resource | Designation | Source or reference | Identifiers | Additional information |
|---|---|---|---|---|
| Antibody | anti-SERBP1 (mouse monoclonal) | Santa Cruz Biotechnology | Cat#:sc100800 | WB (1:100 for human, 1:1000 for cell line samples), IF (1:50 for human, 1:200 for cell line samples), PLA (1:100) |
| Antibody | anti-GAPDH (rabbit polyclonal) | Abcam | Cat#:ab9485 | WB (1:1000) |
| Antibody | anti-G3BP1 (rabbit polyclonal) | Cell Signaling | Cat#:17798 S | IF (1:200) |
| Antibody | anti-PARP1 (rabbit monoclonal) | Cell Signaling | Cat#:9532 S | WB (1:1000), PLA (1:200) |
| Antibody | anti-SYNCRIP (rabbit polyclonal) | Invitrogen | Cat#:PA5-59501 | WB (1:1000) |
| Antibody | anti-hnRNPU (rabbit monoclonal) | Cell Signaling | Cat#:34095 | WB (1:1000) |
| Antibody | anti-Puromycin (mouse monoclonal) | Kerafast | Cat#:EQ0001 | WB (1:1000) |
| Antibody | anti-His (mouse monoclonal) | Santa Cruz Biotechnology | Cat#:sc-53073 | WB (1:1000) |
| Antibody | anti-Flag (mouse monoclonal) | Invitrogen | Cat#:MA1-91878 | WB (1:1000) |
| Antibody | anti-GAPDH (mouse monoclonal) | Santa Cruz Biotechnology | Cat#:SC32233 | WB (1:2000) |
| Antibody | anti-β-Actin (rabbit polyclonal) | Abcam | Cat#:ab8227 | WB (1:1000) |
| Antibody | anti-β-Tubulin (mouse monoclonal) | Sigma-Aldrich | Cat#:T8328 | WB (1:2000) |
| Antibody | anti-α-Tubulin (mouse monoclonal) | Invitrogen | Cat#:236–10501 | IF (1:200) |
| Antibody | anti-FBL (rabbit monoclonal) | Cell Signaling | Cat#:C13C3 | WB (1:1000) |
| Antibody | anti-NCL (rabbit monoclonal) | Cell Signaling | Cat#:D4C70 | WB (1:1000) |
| Antibody | anti-G3BP1 (mouse monoclonal) | Abcam | Cat#:ab56574 | IF (1:200) |
| Antibody | Anti-phospho-Tau (Thr231) (mouse monoclonal) | Thermo Fisher | Cat#:MN1040 | IF(1:250), PLA(1:500) |
| Antibody | HRP-conjugated anti-rabbit (goat polyconal) | Santa Cruz Biotechnology | Cat#:sc-2030 | WB (1:5000) |
| Antibody | HRP-conjugated anti-mouse (goat, polyclonal) | Santa Cruz Biotechnology | Cat#:sc-2005 | WB (1:5000) |
| Antibody | Alexa Fluor 488-conjugated anti-rabbit (goat polyclonal) | Invitrogen | Cat#:A11008 | IF (1:500) |
| Antibody | Alexa Fluor 568-conjugated anti-mouse (goat polyclonal) | Invitrogen | Cat#:A11004 | IF (1:500) |
| Antibody | Anti-poly-ADP-ribose binding reagent (with rabbit Fc-tag) | Millipore | Cat#:MABE1031; RRID:AB_2665467 | WB (1:1000) |
| Recombinant DNA reagent | pEF1 (plasmid) | Thermo Fisher | Cat#:V92120 | |
| Recombinant DNA reagent | pSBP-SERBP1 (plasmid) | This paper | | SERBP1 ORF and SBP-tag cloned in frame in pEF1 backbone |
| Recombinant DNA reagent | pUltra-SERBP1 lentiviral vector | *Kosti et al., 2020* | | control for pulldown experiments (no SBP-tag) |
| Recombinant DNA reagent | pcDNA3.1-mGreenLantern (plasmid) | Addgene | RRID:Addgene_161912 | |

*Continued on next page*

*Continued*

| Reagent type (species) or resource | Designation | Source or reference | Identifiers | Additional information |
|---|---|---|---|---|
| Recombinant DNA reagent | mGreen-SERBP1 (plasmid) | This paper | | SERBP1 ORF cloned in frame in pcDNA3.1-mGreenLantern backbone |
| Recombinant DNA reagent | Flag-PARP1 | Addgene | RRID:Addgene_111575 | |
| Peptide, recombinant protein | 6xHis-tagged SERBP1 | *Baudin et al., 2021* | | |
| Commercial assay or kit | Lipofectamine RNAiMAX | Invitrogen | Cat#:13778150 | |
| Commercial assay or kit | Streptavidin beads | GE Healthcare Life Sciences | Cat#:17-5113-01 | |
| Commercial assay or kit | Cell Titer Glo 2.0 | Promega | Cat#:G9243 | |
| Commercial assay or kit | Duolink PLA in Situ Red starter kit mouse/rabbit | Sigma-Aldrich | Cat#:DUO92101 | |
| Chemical compound, drug | Cambridge Cancer Compound Library | Selleck Chem | Cat#:L2300 | 100 nM in 0.1% DMSO treatment concentration |
| Chemical compound, drug | Puromycin | Sigma-Aldrich | Cat#:P7255 | |
| Chemical compound, drug | Paclitaxel | Cayman Chem | Cat#:10461 | |
| Chemical compound, drug | PARP inhibitor PJ34 | Enzo | Cat#:ALX-270–289 | |
| Software, algorithm | Mascot v2.7.0 | Matrix Science | RRID:SCR_014322 | |
| Software, algorithm | Scaffold v4.9.0 | Proteome Software | RRID:SCR_014321 | |
| Software, algorithm | ImageJ FIJI | NIH | RRID:SCR_002285 | |
| Software, algorithm | BioInfoRx | https://bioinforx.com/apps/venn.php | | |
| Software, algorithm | Nematode genome comparison browser | http://nemates.org/MA/progs/overlap_stats.html | | |
| Software, algorithm | ShinyGO v0.67 & v0.77 | *Ge et al., 2020* | | |
| Software, algorithm | Metascape v3.5 | *Zhou et al., 2019* | | |
| Software, algorithm | Panther v17.0 | *Thomas et al., 2022* | | |
| Software, algorithm | Revigo | *Supek et al., 2011* | | |
| Software, algorithm | STRING v11.5 | *Szklarczyk et al., 2023* | | |
| Software, algorithm | Cytoscape | *Doncheva et al., 2023* | | |
| Software, algorithm | R2 | http://r2.amc.nl | | |
| Software, algorithm | dcGO Enrichment mining service | *Fang and Gough, 2013* | | |
| Software, algorithm | STAR v2.7.7.a | *Dobin et al., 2013* | | |
| Software, algorithm | rMATS v4.1.2 | *Shen et al., 2014* | | |
| Software, algorithm | rmats2sashimiplot tool | https://github.com/Xinglab/rmats2sashimiplot; *Xinglab, 2024* | | |

*Continued on next page*

*Continued*

| Reagent type (species) or resource | Designation | Source or reference | Identifiers | Additional information |
|---|---|---|---|---|
| Software, algorithm | BEDTools intersect software | *Quinlan and Hall, 2010* | | |
| Software, algorithm | Kallisto v0.46.1 | *Bray et al., 2016* | | |
| Software, algorithm | R package tximport | *Soneson et al., 2015* | | |
| Software, algorithm | DESeq2 | *Love et al., 2014* | | |
| Other | Fluor Save | Invitrogen | Cat#:345789 | Reagent, microscopy |
| Other | Prolong Gold Antifade with DAPI | Thermo Fisher | Cat#:P36931 | Reagent, microscopy |

## Cell culture, transfection, knockdown

U251 and U343 glioblastoma cell lines were obtained from Uppsala University (Uppsala, Sweden). 293T cells were obtained from the American Type Tissue Collection (ATCC). All cell lines were cultured in DMEM medium (HyClone, Cat# SH30243.01) supplemented with 10% FBS (Corning, Cat# 35015CV) and 1% penicillin/streptomycin (Gibco, Cat# 10378016). Cells were maintained at 37 °C in a 5% $CO_2$ atmosphere. Cell lines were confirmed to be correct using Short Tandem Repeat profiling. They were regularly tested for mycoplasma contamination.

Knockdown of SERBP1 (siSERBP1 KD) in U251 and U343 cells was performed by reverse transfection with 25 nM of control or SERBP1 SMARTpool siRNA (Dharmacon, Cat# L-020528-01-0005) using Lipofectamine RNAiMAX (Invitrogen, Cat# 13778150).

## Plasmid constructs

A vector containing the Streptavidin-Binding Peptide (SBP)-Tag was prepared in pEF1 (Thermo Fisher, Cat# V 92120), and SERBP1 ORF was cloned in frame with the His-Myc-tag present in the vector to create pSBP-SERBP1. The SBP-tag contains 38 amino acids and binds to streptavidin with an equilibrium dissociation constant of 2.5 nM (*Keefe et al., 2001*). Ultra-SERBP1 lentiviral vector (*Kosti et al., 2020*) was used as a control in pulldown experiments. SERBP1 ORF was inserted into the pcDNA3.1-mGreenLantern plasmid (RRID:Addgene_1619122) to make mGreen-SERBP1. Flag-PARP1 was obtained from Addgene (RRID:Addgene_111575).

| Primer name | sequence |
|---|---|
| SERBP1_for_ EcoRI | tatagaattcATGCCTGGGCACTTAATGCCTGGGCACTTA |
| SERBP1_rev_XbaI | tatatctagaAGCCAGAGCTGGGA |
| mGL_SERBP1_BsrGI | actgtacaagATGCCTGGGCACTTACAG |
| mGL_SERBP1_XbaI | atctagattaAGCCAGAGCTGGGAA |

## Identification of SERBP1 interactors

293T cells were seeded (triplicate) in 10 cm dishes the day before transfection. pSBP-SERBP1 or pUltra-SERBP1(control) plasmids were transfected using $CaCl_2$. 48 hr later, cells were collected and washed twice in cold PBS. Total protein extract was prepared by using TNE buffer (Tris pH 7.4 10 mM, NP-40 1%, NaCl 150 mM, and phosphatase inhibitor cocktails, Thermo Fisher Scientific, Cat# 78430), incubating cells for 30 min on ice, and then sonicating five times for 3 s at 20% amplitude and 10 s intervals. A second batch of samples was prepared by using polysomal lysis buffer (KCl 100 mM, EDTA 25 mM, $MgCl_2$ 5 mM, HEPES pH 7.0 10 mM, NP-40 0.5%, DTT 2 mM, VRC 0.4 mM, glycerol 10%, and phosphatase inhibitor cocktails, Thermo Fisher Scientific). Cell lysates were centrifuged at 4 °C to eliminate debris and supernatants were collected and used later in pulldown assays.

50 µl of packed streptavidin beads (GE Healthcare Life Sciences, Cat# 17-5113-01) were washed two times in PBS and subsequently blocked in 1% BSA/PBS for 30 min at 4 °C. Beads were finally washed five times in TNE buffer and then combined with cell extracts. The solution was incubated at room temperature for 2 hr and beads were later recovered via centrifugation at 4 °C. The supernatant was discarded, and beads were washed five times with 1 ml of TNE buffer. To elute proteins, samples

were resuspended in elution buffer (Tris pH 7.4 50 mM, NaCl 250 mM, NP-40 0.5%, deoxycholate 0.1%, 10 mM biotin), mixed, and incubated for 45 min at 37 °C in a thermo-shaker. The supernatant was collected after centrifugation for mass spectrometry analysis.

## Protein identification by mass spectrometry

### Analysis of proteins associated with SERBP1

Proteins were separated by 1D SDS-PAGE using a Criterion XT 12% gel that was electrophoresed for ~1.5 cm and then stained with Coomassie blue (one experiment per gel). The protein-containing region of each gel lane was divided into six slices which were individually reduced in situ with TCEP [tris(2-carboxyethyl)phosphine] and alkylated in the dark with iodoacetamide prior to treatment with trypsin. Each digest was analyzed by capillary HPLC-electrospray ionization tandem mass spectrometry on a Thermo Scientific Orbitrap Fusion Lumos mass spectrometer. On-line HPLC separation was accomplished with an RSLC NANO HPLC system (Thermo Scientific/Dionex): column, PicoFrit (New Objective; 75 µm i.d.) packed to 15 cm with C18 adsorbent (Vydac; 218 MS 5 µm, 300 Å). Precursor ions were acquired in the Orbitrap mass spectrometer in centroid mode at 120,000 resolution (*m/z* 200); data-dependent higher-energy C-trap dissociation (HCD) spectra were acquired at the same time in the linear trap using the 'top speed' option (30% normalized collision energy). Mascot (v2.7.0; Matrix Science) was used to search the spectra against a combination of the human subset of the UniProt database plus a database of common contaminants [UniProt_Human 20181204 (95,936 sequences; 38,067,061 residues); contaminants 20120713 (247 sequences; 128,130 residues)]. Subset search of the identified proteins by X! Tandem, cross-correlation with the Mascot results, and determination of protein and peptide identity probabilities were accomplished by Scaffold (v4.9.0; Proteome Software). The thresholds for acceptance of peptide and protein assignments in Scaffold were set to yield <1% protein FDR. Corresponding UniProt IDs for the proteins were obtained using the UniProt ID mapping service (*Bateman et al., 2023*).

## Western blots

### Cells

Cells were harvested and lysed in Laemmli sample buffer (BioRad, Cat# 1610737). Cell extracts were separated on an SDS-PAGE gel, and transferred to PVDF membranes, previously activated with methanol. Membranes were blocked in 5% milk Tris-buffered saline with Tween 20 and probed with a collection of different antibodies listed below. Horseradish peroxidase (HRP)-conjugated goat anti-rabbit antibody (Santa Cruz Biotechnology, Cat# sc-2030) or HRP-conjugated goat anti-mouse (Santa Cruz Biotechnology, Cat# sc-2005) were used as secondary antibodies. Immobilon Western chemoluminescence substrate (Millipore, Cat# WBKLS0500) was used to detect selected proteins.

### Human samples

Western blots were performed with proteins extracted from control and Alzheimer's brains. Approximately 10 µg of protein were loaded into precast NuPAGE 4–12% Bis-Tris gels (Invitrogen) for analysis by sodium dodecyl sulfate–polyacrylamide gel electrophoresis. The separated proteins were subsequently transferred onto nitrocellulose membranes and blocked overnight at 4 °C with 10% nonfat dry milk. The membranes were then probed for 1 hr at room temperature with primary antibodies - anti-SERBP1 (1:100, sc100800) and anti-glyceraldehyde 3-phosphate dehydrogenase (GAPDH; 1:1000, ab9485, Abcam) - diluted in 5% nonfat dry milk. SERBP1 immunoreactivity was detected with a horseradish peroxidase–conjugated anti-mouse IgG (1:5000, GE Healthcare). GAPDH immunoreactivity was detected using an anti-mouse IgG (1:6000, GE Healthcare) diluted in 5% nonfat dry milk. An enhanced chemiluminescence reagent (ECL Plus, Amersham) was used to visualize the bands. The amount of protein was normalized and quantified using the loading control GAPDH.

## Antibodies

anti-SERBP1 (Santa Cruz Biotechnology, Cat# SC100800), anti-G3BP1 (Cell Signaling, Cat# 17798 S), anti-PARP1 (Cell Signaling, Cat# 9532 S), anti-SYNCRIP (Invitrogen, Cat# PA5-59501), anti-hnRNPU (Cell Signaling, Cat# 34095), anti-Puromycin (Kerafast, Cat# EQ0001), anti-His (Santa Cruz Biotechnology, Cat# sc-53073), anti-Flag (Invitrogen, Cat# MA1-91878), anti-GAPDH (Santa Cruz Biotechnology, Cat# SC32233), anti-β-Actin (Abcam, Cat# ab8227), anti-β-Tubulin (Sigma-Aldrich, Cat#

T8328), anti-α-Tubulin (Invitrogen, Cat# 236–10501), FBL (Cell Signaling, Cat# C13C3), NCL (Cell Signaling, Cat# D4C70).

## High-throughput drug screening

High-throughput screening was conducted with the Cambridge Cancer Compound Library (Selleck Chem, Cat# L2300) of 293 compounds of interest (*Supplementary file 7*). All compounds dissolved in DMSO to a stock concentration of 10 mM and further diluted in PBS to achieve a final treatment concentration of 100 nM and a final DMSO concentration of 0.1%. Four replicates of both U343 OE and U343 Control cells were plated in 384 well white polystyrene microplates (Thermo Fisher Scientific) at a seeding density of $5.0 \times 10^3$ cells/well 12 hr before treatment. Cell seeding and treatment was conducted by an automated electronic pipetting system to increase pipetting accuracy and to minimize variability between replicates (Integra VIAFLO 384). After 24 hr of treatment, cells were incubated with Cell Titer Glo 2.0 (Promega, Cat# G9243) according to the manufacturer's protocol. The addition of the reagent lyses cells and attaches a luminescent tag to ATP molecules in order to assess cell viability. Within 10 min of Cell Titer Glo 2.0 reagent addition, ATP measurements were then collected as luminescence intensities by a Tecan Spark microplate reader. Dixon's Q test was used to eliminate outliers prior to statistical analysis. Experimental ATP intensities were normalized to respective vehicle controls to assess cell viability following drug treatment. Student's Ttest was used to evaluate significant differences between reactions of U343 and U343 SERBP1 OE to each treatment.

## Puromycin assays

siSERBP1 KD in U251 and U343 cells was performed as described above. 72 hr after transfection, cells were incubated with 10 µg/ml Puromycin (Sigma-Aldrich, Cat# P7255) for 15 min at 37 °C and 5% $CO_2$. Cells were then placed on ice and washed twice with cold PBS before being harvested in Laemmli sample buffer. Western blots were performed as described above with 15 µg protein extracts.

## Protein co-localization

U251 cells were cultured on glass coverslips until 80% confluent and fixed in 4% paraformaldehyde followed by permeabilization with 0.1% TritonX-100 for 2 min. Cells were then stained with primary antibody for 1 hr followed by the secondary antibody conjugated to Alexa Fluor 488 or 568 (Invitrogen, Invitrogen, Cat# A11004 and A11008). After washing with PBS, cells were stained with DAPI and mounted onto glass slides with FluorSave (Invitrogen, Cat# 345789) before being analyzed by fluorescence microscopy.

## Impact of SERBP1 on cell division

24 hr after siSERBP1 KD in U251 and U343, cells were exposed to 20 nM of Paclitaxel (Cayman Chem, Cat# 10461) for 24 hr. Cells were fixed in 4% PFA and stained with α-Tubulin and DAPI. The experiment was performed in triplicates and 1000 cells were counted per replicate. Statistical significance was evaluated by Student's t-test.

## In vitro PARylation assay

Flag-tagged PARP1 was expressed in insect cells and purified as previously described (*Huang et al., 2020*). 6xHis-tagged SERBP1 was purified as in *Baudin et al., 2021*. Briefly, purified recombinant PARP1 (0.1 µM) and SERBP1 (0.6 µM) were combined in an ADP-ribosylation reaction (20 µL) with or without sheared salmon sperm DNA (sssDNA; 100 ng/µL) and NAD+ (100 µM). The reactions were incubated at room temperature for 15 min. The reaction products were analyzed on 8% PAGE-SDS gels with silver staining or Western blotting for PAR using a recombinant anti-poly-ADP-ribose binding reagent (Millipore, Cat# MABE1031; RRID:AB_2665467).

## SERBP1 impact on PARylation

pcDNA3.1-EF1a-mGreenLantern-SERBP1 or control plasmid pcDNA3.1-EF1a-mGreenLantern was transfected into 293T cells. 48 hr later, cells were collected and lysed in Laemmli sample buffer. 10 µg of extracted proteins were separated on SDS-PAGE gel and Western blot was performed as described. Poly-ADP-ribose binding reagent (Sigma-Aldrich, Cat# MABE1031) was used to detect oligo- and poly-ADP-ribosylated (PARylated) proteins.

## Sensitivity to PARP inhibitor

U251 and U343 siSERBP1 knockdown was performed as described above. 5 µM of PARP inhibitor PJ34 (Enzo, Cat# ALX-270–289) was used to treat the cells 24 hr later. The Essen Bioscience IncuCyte automated microscope system was used to follow the proliferation of U251 and U343 glioblastoma cells over 150 hr. Differences in proliferation between single and combined treatment at 100 or 150 hr were analyzed graphically. Statistical significance was calculated by Student's t-test and the combination treatment was evaluated with the Combination Index (*Chou, 2010*).

## Immunolabeling of fixed human brain sections

Frozen cortical sections were first fixed in chilled 20% methanol. To eliminate autofluorescence from lipofuscin, we incubated the sections with TrueBlack Lipofuscin Autofluorescence Quencher (#23007, Biotium) for 10 min at room temperature. After three washes with ethanol 70%, the sections were permeabilized in PBS/0.4% Triton-X100 for 5 min. After 1 hr of blocking in PBS/10% NGS/0.2% Triton at room temperature, the sections were incubated in primary antibodies diluted in PBS/10% NGS overnight at 4 °C. Immunolabeling for G3BP1 (5 µg/ml, 2F3, Abcam Cat# ab56574), Phospho-Tau (Thr231) (1:500, AT180, Thermo Fisher Scientific #MN1040) and SERBP1 (1:50, 1B9, Santa Cruz Bio Cat# sc-100800) was performed in all control (N=3) and AD (N=3, Braak Stage V-VI) human brains. The next day, sections were washed three times in PBS (5 min each), and secondary antibodies were applied for 45 min: Alexa Fluor 488 and 568 (1:200, Life Technologies). After further washing in PBS, slides were then mounted with Prolong Gold Antifade with DAPI (Thermo Fisher Scientific Cat# P36931) to stain the nuclei. Granule density of G3BP1 and SERBP1 were measured in 3 Ctr and 3 AD cases, and images in triplicate from each case were analyzed by ImageJ FIJI (NIH) software. SERBP1 associated with stress granules was measured as the ratio between colocalizing SERBP1/G3BP1 puncta and G3BP1 total granules in the cortical section using ImageJ FIJI (NIH) software.

## Human tissue harvesting

Frontal cortices of frozen brain tissues from age-matched control subjects (N=6), AD cases (N=6) were received as frozen blocks from the Institute for Brain Aging and Dementia at UC Irvine, approved by the Institutional Ethics Committee. Brain tissues were homogenized in 1 X PBS mixed with a protease inhibitor cocktail (Roche) and phosphatase inhibitor (Sigma) at 1:3 (w/v) dilution of brain: PBS. Samples were then centrifuged at 10,000 rpm for 20 min at 4 °C. The supernatants, PBS-soluble fractions were aliquoted, snap-frozen, and stored at −80 °C until use. The pellets were resuspended in the homogenization buffer (1 X PBS) and were considered as insoluble fractions. They were also aliquoted and

**Table 1.** Brain tissues analyzed in this study from diseased and age-matched non-demented control subjects are summarized.

| Clinical Diagnosis | Case Number | Age | Gender | PMI (Hours) | Braak Tangles (0–6) | Application |
|---|---|---|---|---|---|---|
| AD | 1154 | 86 | M | 3.25 | 6 | IF/PLA |
| AD | 1098 | 81 | F | 2.75 | 5 | IF/PLA |
| AD | 5773 | 74 | M | 10 | 5 | IF/PLA |
| AD | 5779 | 73 | M | 15 | 6 | WB |
| AD | 5781 | 88 | M | 10 | 5 | WB |
| AD | 5829 | 68 | M | 12 | 6 | WB |
| Control | 5263 | 88 | M | 12.17 | 1 | IF/PLA |
| Control | 1161 | 84 | F | 2.50 | 0 | IF/PLA |
| Control | 1106 | 79 | M | 1.75 | 2 | IF/PLA |
| Control | 1796 | 81 | M | 8 | 0 | WB |
| Control | 2–99 | 74 | F | 2.8 | 2 | WB |
| Control | 13–01 | 95 | M | 3.7 | 1 | WB |

IF: Immunofluorescence; PLA: Proximity Ligation Assay; PMI:Postmortem Interval; WB:Western Blotting.

frozen at –80 °C until use. Brain tissues analyzed in this study from diseased and age-matched non-demented control subjects are listed in *Table 1*.

## Proximity ligation assay (PLA)

Detection of protein-protein interactions was conducted by an in-situ proximity ligation assay. The method depends on the recognition of target molecules in close proximity (<40 nm) by pairs of affinity probes, giving rise to an amplifiable detection signal. Briefly, PLA in mouse brain tissue has been performed using Duolink PLA in Situ Red starter kit mouse/rabbit (Sigma-Aldrich, DUO92101). The staining was performed following each protocol passage without modifications. Time of incubation and concentration of antibodies is established from IF protocol. Primary antibodies used for in situ proximity assay were: SERBP1 (1:100), PARP1 (1:200), and pThr231-Tau (1:500). Amplified red signal has been detected using Keyence Microscope.

## Venn diagram statistics

Venn diagrams were created using online software (https://bioinforx.com/apps/venn.php). P-values for overlaps between two groups were calculated with a separate online tool (http://nemates.org/MA/progs/overlap_stats.html).

## Gene ontology and network analyses

Gene Ontology (GO) enrichment analyses were performed using ShinyGO (versions 0.67 and 0.77), Metascape (version 3.5), and Panther (version 17.0; *Ge et al., 2020*; *Zhou et al., 2019*; *Thomas et al., 2022*). Associations between GO terms were established using Revigo (*Supek et al., 2011*). Network analyses were conducted using STRING (version 11.5) and Metascape (*Zhou et al., 2019*; *Szklarczyk et al., 2023*) and visualized using Cytoscape (version 3.9.1; *Doncheva et al., 2023*).

## Expression correlation analyses

We performed expression correlation analyses using resources in R2 (*Koster, 2023*) to identify genes with a strong positive correlation with SERBP1 ($R \geq 0.3$, p-value $\leq 0.05$, Pearson correlation) in the TCGA glioblastoma set (RNA-Seq samples), sarcoma (TCGA), bladder urothelial carcinoma (TCGA), neuroblastoma (SEQC), pancreatic adenocarcinoma (TCGA), and normal brain (Hardy, $R \geq 0.5$ was selected in this case due to the high number of correlated genes). Additional analyses were conducted using the Ivy GAP glioblastoma dataset *Puchalski et al., 2018* following their default parameters ($R \geq 0.3$).

The Cortecon resource (*van de Leemput et al., 2014*) was used to identify genes that share similarity with SERBP1 regarding their expression profile during in vitro cortex development. SERBP1 is part of cluster 5 which is linked to pluripotency. Genes that were in clusters related to pluripotency and whose overall profile resembles cluster 5 were selected for further analyses.

## Identification of protein features and domain

Pfam accession numbers were obtained using the UniProt ID mapping service (*Bateman et al., 2023*). R code was used to count the frequency of each accession number and create the consolidated data with fold enrichment and fdr-values. All accession numbers with a count above the average were analyzed with the dcGO Enrichment mining service (*Fang and Gough, 2013*).

## Databases and datasets used for characterization of SERBP1-associated proteins

STM1 interactors were downloaded from BioGRID (version 4.4) *Stark et al., 2006* and human orthologs were obtained using the WORMHOLE website (*Sutphin et al., 2016*). Proximity-dependent biotinylation (Bio-ID) data for SERBP1 were taken from *Go et al., 2021*; *Youn et al., 2018*. Proteins of the SARS-CoV-RNA interactome were derived from *Lee et al., 2021* and those of the PARP1 interactome from *Mosler et al., 2022*. RGG box-containing proteins were obtained from *Thandapani et al., 2013*, and RRM motif-containing proteins from UniProt (*Bateman et al., 2023*). G4 binding proteins were extracted from *Su et al., 2021b*; *Herviou et al., 2020*, PAR binding proteins were obtained from *Dasovich et al., 2021*, and PARylated proteins were identified by high-throughput studies described in *Gibson et al., 2016*; *Martello et al., 2016*. The MobiDB database (version 5.0; *Piovesan et al.,*

*2023*) was used for identifying IDR-containing proteins. Data on the Tau aggregate interactome were derived from different studies compiled by *Kavanagh et al., 2022*.

## Splicing analyses

To identify splicing alterations induced by SERBP1 and hnRNPU knockdown, adapter sequences were first trimmed from raw RNA-Seq reads of control and knockdown samples (*Kosti et al., 2020*). RNA-Seq reads were then aligned against the human reference genome (version GRCh38; *Kitts et al., 2016*) and matching GENCODE transcriptome (v29; *Frankish et al., 2021*) using STAR (default parameters; version 2.7.7 .a; *Dobin et al., 2013*). High-quality mapped reads (q>20) from control and knockdown samples were processed using rMATS (default parameters; version v4.1.2; *Shen et al., 2014*) to characterize splicing events. Events were classified into exon skipping (SE), mutually exclusive exons (MXE), intron retention (RI), alternative donor site (A5SS) or alternative acceptor site (A3SS). Splicing events were considered significant if their FDR-adjusted p-values were below 0.05 and absolute delta Percent Splice Inclusion (deltaPSI) above 0.1. Sashimi plots of splicing events were created using the rmats2sashimiplot tool https://github.com/Xinglab/rmats2sashimiplot (*Xinglab, 2024*). To assess SERBP1 binding to the identified splicing alterations, we obtained processed iCLIP data from *Martini et al., 2021*. The presence of iCLIP sites in the splicing alterations induced by SERBP1 knockdown was evaluated considering a window of 100 nt around splicing events using the bedtools intersect software (*Quinlan and Hall, 2010*).

## Identification of snoRNAs and scaRNAs bound by SERBP1

RIP-sequencing results for SERBP1 were obtained from *Kosti et al., 2020*. Sequencing reads were processed using Kallisto (default parameters, version 0.46.1; *Bray et al., 2016*) with an index of 31 k-mers and GENCODE (v28) as the reference for the human transcriptome (*Frankish et al., 2021*). Transcript abundance estimates were then collapsed into gene-level counts using the R package tximport (*Soneson et al., 2015*). Differential gene expression analyses were performed using DESeq2 (*Love et al., 2014*). We compared SBP-SERBP1 versus SBP-GST samples and used log2 fold change ≥0.5 and adjusted (false discovery rate, FDR) p-value <0.05 to identify snoRNAs and scaRNAs preferentially associated with SERBP1.

## Acknowledgements

Mass spectrometry analyses were conducted at the Institutional Mass Spectrometry Laboratory, University of Texas Health Science Center, supported in part by NIH grant P30 CA54174-23 (S.T. Weintraub, Mays Cancer Center Mass Spectrometry Shared Resource) and the University of Texas System Proteomics Core Network for purchase of the Orbitrap Fusion Lumos mass spectrometer, with expert technical assistance of Sammy Pardo and Dana Molleur. We would like to thank Dr. Jernej Ule and Dr. Rupert Faraway for sharing the SERBP1 CLIP data and their valuable input. The authors would like to acknowledge Karen Klein for excellent editorial service. This work was supported by the Cancer Prevention and Research Institute of Texas [RP200595 to LP], Alzheimer's Association [AARGD-NTF-22-968603 to LP] and a Mays Cancer Center Pilot Grant [to LP]. PAFG was supported by grants from Serrapilheira Foundation and Fundação de Amparo à Pesquisa do Estado de São Paulo (FAPESP) [2018/15579-8]. MM was sponsored by the Alzheimer's Association [AARF-21-720991]. GDAG was supported by a fellowship from FAPESP [2017/19541-2] and Young Scientist program, Hospital Sírio-Libanês. KB and ML were sponsored by the The German Academic Exchange Service (DAAD). AK was supported by NIH Supplement 2R01 HG006015 and the Greehey Foundation. DSL was partially supported by a Voelker Fund Young Investigator Grant.

## Additional information

### Funding

| Funder | Grant reference number | Author |
|---|---|---|
| Cancer Prevention and Research Institute of Texas | RP200595 | Luiz O Penalva |
| Alzheimer's Association | AARGD-NTF-22-968603 | Luiz O Penalva |
| Mays Cancer Center | Pilot Grant | Luiz O Penalva |
| Serrapilheira Foundation | | Pedro AF Galante |
| Fundação de Amparo à Pesquisa do Estado de São Paulo | 2018/15579-8 | Pedro AF Galante |
| Alzheimer's Association | AARF-21-720991 | Mauro Montalbano |
| Fundação de Amparo à Pesquisa do Estado de São Paulo | 2017/19541-2 | Gabriela DA Guardia |
| Hospital Sírio-Libanês | | Gabriela DA Guardia |
| Deutscher Akademischer Austauschdienst | | Kira Breunig Mujia Li |
| National Institutes of Health | 2R01 HG006015 | Adam Kosti Luiz O Penalva |
| Greehey Foundation | | Adam Kosti |
| Voelker Fund | Young Investigator Grant | David S Libich |
| National Institutes of Health | CA54174-23 | Adam Kosti |

The funders had no role in study design, data collection and interpretation, or the decision to submit the work for publication.

### Author contributions

Kira Breunig, Conceptualization, Data curation, Formal analysis, Validation, Investigation, Methodology, Writing – original draft, Writing – review and editing; Xuifen Lei, Data curation, Formal analysis, Supervision, Validation, Investigation, Methodology, Writing – original draft, Writing – review and editing; Mauro Montalbano, Conceptualization, Data curation, Formal analysis, Validation, Visualization, Methodology, Writing – original draft, Writing – review and editing; Gabriela DA Guardia, Data curation, Formal analysis, Validation, Investigation, Visualization, Methodology, Writing – original draft, Writing – review and editing; Shiva Ostadrahimi, Formal analysis, Investigation, Methodology, Writing – original draft; Victoria Alers, Formal analysis, Investigation, Visualization, Writing – original draft; Adam Kosti, Data curation, Formal analysis, Investigation, Writing – original draft; Jennifer Chiou, Lily Wang, Mujia Li, Formal analysis, Investigation; Nicole Klein, Corina Vinarov, Formal analysis, Investigation, Visualization; Weidan Song, Data curation, Formal analysis, Investigation, Visualization, Methodology, Writing – original draft; W Lee Kraus, Data curation, Supervision, Investigation; David S Libich, Supervision, Investigation; Stefano Tiziani, Conceptualization, Investigation, Visualization, Methodology, Writing – original draft, Writing – review and editing; Susan T Weintraub, Conceptualization, Data curation, Formal analysis, Methodology, Writing – original draft, Writing – review and editing; Pedro AF Galante, Conceptualization, Resources, Data curation, Formal analysis, Supervision, Investigation, Visualization, Writing – original draft, Writing – review and editing; Luiz O Penalva, Conceptualization, Resources, Data curation, Formal analysis, Supervision, Funding acquisition, Investigation, Visualization, Writing – original draft, Project administration, Writing – review and editing

### Author ORCIDs

Xuifen Lei ⓘ https://orcid.org/0009-0000-1071-6190
Mauro Montalbano ⓘ https://orcid.org/0000-0002-0456-452X
Gabriela DA Guardia ⓘ https://orcid.org/0000-0002-1789-2768

W Lee Kraus ⓘ https://orcid.org/0000-0002-8786-2986
David S Libich ⓘ https://orcid.org/0000-0001-6492-2803
Susan T Weintraub ⓘ https://orcid.org/0000-0002-8328-7814
Pedro AF Galante ⓘ https://orcid.org/0000-0002-4820-4155
Luiz O Penalva ⓘ https://orcid.org/0000-0003-4491-6769

Reviewer #1 (Public review): https://doi.org/10.7554/eLife.98152.3.sa1
Reviewer #2 (Public review): https://doi.org/10.7554/eLife.98152.3.sa2
Author response https://doi.org/10.7554/eLife.98152.3.sa3

## Additional files

### Supplementary files

Supplementary file 1. SERBP1-associated proteins identified by pulldown in 293T cells. Mass spectrometry results of pulled-down proteins with respective counts in SBP-SERBP1 and control cells in two different experimental conditions. The summary sheet displays gene names and UniProt IDs (*Bateman et al., 2023*) of identified SERBP1 interactors. Additional sheets show SERBP1-associated proteins present in the nucleus and nucleolus. Also included are complete results of ShinyGO (*Ge et al., 2020*) and Metascape (*Zhou et al., 2019*) GO enrichment analyses of SERBP1 and all its identified associated proteins as well as comparisons of newly identified SERBP1 interactors with previous proximity-label (BIO-ID) studies (*Go et al., 2021*; *Youn et al., 2018*) and compilations from BioGRID (*Stark et al., 2006*).

Supplementary file 2. snoRNAs and scaRNAs bound by SERBP1 according to RIP-Seq analysis (*Kosti et al., 2020*) and comparisons to PARP1 bound snoRNAs identified CLIP-identified (*Mekishvili et al., 2017*).

Supplementary file 3. BioGRID data for STM1, SERBP1 yeast homolog. STM1's protein interactors as curated by BioGRID (*Stark et al., 2006*) with their gene ontologies (*Ge et al., 2020*), human homologs as derived from WORMHOLE (*Sutphin et al., 2016*), and overlap with SERBP1-associated proteins.

Supplementary file 4. Protein domains and intrinsically disordered regions of SERBP1-associated proteins. Analysis of protein domains present in SERBP1-associated proteins (*Bateman et al., 2023*) with their respective counts in reference to the total human proteome (*Sonnhammer et al., 1997*). Additional sheets display the results of gene ontology analyses for top-represented domains (*Fang and Gough, 2013*). Also shown are derived and curated intrinsically disordered proteins associated with SERBP1 according to MobiDB (*Piovesan et al., 2023*).

Supplementary file 5. Gene expression correlation analysis. Significant expression correlation values for SERBP1 and associated proteins in glioblastoma, neuroblastoma, pancreatic adenocarcinoma, bladder urothelial carcinoma, sarcoma, and normal brain datasets (*Koster, 2023*; *Puchalski et al., 2018*). Additionally, a list of SERBP1-associated proteins displaying a similar expression profile during cortex development according to Cortecon (*van de Leemput et al., 2014*). The last sheet shows the top SERBP1 interactors with the highest number of high expression correlation instances and their respective proteomics counts from our pulldown experiment.

Supplementary file 6. Characteristics of SERBP1-associated proteins. SERBP1-associated proteins are organized according to the presence of RGG boxes (*Thandapani et al., 2013*), RRM motifs (*Bateman et al., 2023*), G4 binding (*Su et al., 2021b*; *Herviou et al., 2020*), PAR binding (*Dasovich et al., 2021*), PARylation (*Gibson et al., 2016*; *Martello et al., 2016*) and overlap with SARS-CoV-2 (*Lee et al., 2021*) and PARP1 (*Mosler et al., 2022*) interactomes. Additional sheets show gene ontology results (*Ge et al., 2020*) of SERBP1-associated proteins that (1) get PARylated and/or bind to PAR, (2) overlap with SARS-CoV-2 interactors, or (3) overlap with PARP1 interactors. Mutual SERBP1- and PARP1-associated factors identified in an rRNA transcription and processing screening (*Tafforeau et al., 2013*) are highlighted on the last sheet.

Supplementary file 7. SERBP1 and drug sensitivity. Results of cell viability screening in SERBP1-overexpressing vs. control U343 cells and information on the utilized drugs. Separate sheets list genes conferring high sensitivity to different PARP inhibitors (*Zhang et al., 2023*) and SERBP1 interactors identified in this study.

Supplementary file 8. Impact of SERBP1 and hnRNPU knockdown on the splicing of U251 cells. List of splicing events and respective genes affected in siSERBP1 knockdown vs. control and sihnRNPU

knockdown vs. control U251 cells. Additional files include splicing events affected by SERBP1 knockdown with evidence of SERBP1 binding sites (CLIP sites) (*Kosti et al., 2020*; *Kitts et al., 2016*; *Frankish et al., 2021*; *Dobin et al., 2013*) in the proximity (<100 nt) of regulated splice sites and splicing events affected by both SERBP1 and hnRNPU knockdown in the same direction.

Supplementary file 9. Participation of SERBP1-associated proteins and snoRNAs in membraneless organelles. Proteins associated with SERBP1 and their presence in stress granules, paraspeckles, Cajal bodies, nuclear specks, nucleoli, P-bodies (*Thomas et al., 2022*) and Tau aggregates (*Kavanagh et al., 2022*). UniProt ID mapping (*Bateman et al., 2023*) was used to match gene names and UniProt IDs. SERBP1 interactors implicated in membraneless organelles were evaluated for prevalence of intrinsic disorder (*Piovesan et al., 2023*), G4 binding (*Su et al., 2021b*; *Herviou et al., 2020*), PARylation (*Gibson et al., 2016*; *Martello et al., 2016*), and PAR binding (*Dasovich et al., 2021*). snoRNAs found in a previous study to be enriched in Tau aggregates (*Lester et al., 2021*) that were also identified as SERBP1 targets according to RIP-Seq analysis (*Kosti et al., 2020*).

MDAR checklist

## Data availability

The mass spectrometry datasets generated and analyzed during this study are available in the MassIVE repository as a partner with ProteomeXchange and can be accessed by: (1) MassIVE dataset identifier MSV000091749, ProteomeXchange dataset identifier PXD041664; (2) MassIVE dataset identifier MSV000093355, ProteomeXchange dataset identifier PXD046851. RNAseq datasets are available at the European Nucleotide Archive (ENA) with the accession number PRJEB69681. Code to perform the analysis of SERBP1 impact on splicing can be found in the Open Science Framework.

The following datasets were generated:

| Author(s) | Year | Dataset title | Dataset URL | Database and Identifier |
|---|---|---|---|---|
| Penalva LO | 2024 | SERBP1 interactome defines its novel regulatory roles in the cytoplasm and nucleus in conjunction with PARP1-, G-quadruplex- and PAR-binders | https://massive.ucsd.edu/ProteoSAFe/dataset.jsp?accession=MSV000091749 | MassIVE, MSV000091749 |
| Penalva LO | 2024 | SERBP1 interactome defines its novel regulatory roles in the cytoplasm and nucleus in conjunction with PARP1-, G-quadruplex- and PAR-binders - DIA-MS data | https://massive.ucsd.edu/ProteoSAFe/dataset.jsp?accession=MSV000093355 | MassIVE, MSV000093355 |
| Penalva PO | 2024 | SERBP1 interactome defines its novel regulatory roles in the cytoplasm and nucleus in conjunction with PARP1-, G-quadruplex- and PAR-binders | https://www.ebi.ac.uk/ena/browser/view/PRJEB69681 | EBI European Nucleotide Archive, PRJEB69681 |

The following previously published datasets were used:

| Author(s) | Year | Dataset title | Dataset URL | Database and Identifier |
|---|---|---|---|---|
| Gingras AC | 2018 | Youn_et_al_RNAbodies_SAINT3105_Set1 | https://massive.ucsd.edu/ProteoSAFe/dataset.jsp?accession=MSV000081411 | MassIVE, MSV000081411 |

*Continued on next page*

*Continued*

| Author(s) | Year | Dataset title | Dataset URL | Database and Identifier |
|---|---|---|---|---|
| Gingras AC | 2018 | Youn_et_al_RNAbodies_SAINT3105_Set2 | https://massive.ucsd.edu/ProteoSAFe/dataset.jsp?accession=MSV000081412 | MassIVE, MSV000081412 |
| Gingras AC | 2018 | Youn_et_al_RNAbodies_SAINT3105_Set3 | https://massive.ucsd.edu/ProteoSAFe/dataset.jsp?accession=MSV000081413 | MassIVE, MSV000081413 |
| Gingras AC | 2021 | Go_BioID_humancellmap_HEK293_lowSDS_2019 | https://massive.ucsd.edu/ProteoSAFe/dataset.jsp?accession=MSV000084359 | MassIVE, MSV000084359 |
| Gingras AC | 2021 | Go_BioID_humancellmap_HEK293_highSDS_2019 | https://massive.ucsd.edu/ProteoSAFe/dataset.jsp?accession=MSV000084360 | MassIVE, MSV000084360 |
| Penalva LO | 2020 | RNA sequencing of U251 SERBP1 knockdown cells and RIP sequencing for the identification of SERBP1 targets in 293T cells | https://www.ebi.ac.uk/ena/browser/view/PRJEB35774 | EBI European Nucleotide Archive, PRJEB35774 |
| Kim VN, Choi Y | 2021 | Profile of the SARS-CoV-2 RNA interactome | https://proteomecentral.proteomexchange.org/cgi/GetDataset?ID=PXD024808 | ProteomeXchange, PXD024808 |
| Beli P | 2022 | PARP1 proximity proteomics reveals interaction partners at stressed replication forks | https://proteomecentral.proteomexchange.org/cgi/GetDataset?ID=PXD037154 | ProteomeXchange, PXD037154 |
| Guillonneau F, Millevoi S | 2020 | hnRNP 1 H/F drive RNA G-quadruplex-mediated translation linked to genomic instability and therapy resistance in glioblastoma | https://www.ebi.ac.uk/pride/archive/projects/PXD015609 | PRIDE, PXD015609 |
| Ule J, Parker P | 2021 | SERBP1 iCLIP in mitotic and asynchronous cells with Blu577 treatment | https://www.ebi.ac.uk/biostudies/arrayexpress/studies/E-MTAB-10830 | ArrayExpress, E-MTAB-10830 |

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
