## [Editor Report · eLife Assessment]

This study reports **valuable** insights into the interactome of the RNA-binding protein SERBP1 and possible links through PARylation to diverse processes, including splicing, cell division, and ribosome biogenesis. The diversity of processes SERBP1 may regulate means this work would be of very broad interest to the cell biology community. The proteomics data are **solid**, but the functional connection to downstream processes and the link to Alzheimer's disease, while **compelling**, still require further examination. These latter data currently rely on a very limited set of experiments and patient samples with questionable quality of preservation and methodology.

---

## [Referee Report · Reviewer #1 (Public review)]

Summary:

Here the authors convincingly identify and characterize the SERBP1 interactome and further define its role in the nucleus, where it is associated with complexes involved in splicing, cell division, chromosome structure, and ribosome biogenesis. Many of the SERBP1-associated proteins are RNA-binding proteins and SERBP1 exerts its impact, at least in part, through these players. SERBP1 is mostly disordered but along with its associated proteins displays a preference for G4 binding and can can bind to PAR and be PARylated. They present data that strongly suggest that complexes in which SERBP1 participates are assembled through G4 or PAR binding. The authors suggest that because SERBP1 lacks traditional functional domains yet is clearly involved in distinct regulatory complexes, SERBP1 likely acts in the early steps of assembly through the recognition of interacting sites present in RNA, DNA, and proteins.

Strengths:

The data is very convincing and demonstrated through multiple approaches.

Weaknesses:

None. The authors have adequately addressed earlier reviewer concerns.

---

## [Referee Report · Reviewer #2 (Public review)]

Summary:

In this study the authors have used pull-down experiments in a cell line overexpressing tagged SERPINE1 mRNA binding protein 1 (SERBP1) followed by mass spectrometry-based proteomics, to establish its interactome. Extensive analyses are performed to connect the data to published resources. The authors attempt to connect SERBP1 to stress granules and Alzheimer's disease associated tau pathology. Based on the interactome, the authors propose a cross-talk between SERBP1 and PARP1 functions.

Strengths:

The main strength of this study lies in the extensive proteomics data analysis, and its effort to connect the data to published studies.

Weaknesses:

Support for the proposed model: While the authors propose a feedback regulatory model for SERBP1 and PARP1 function, strong evidence for PARylation modulating SERBP1 functions is lacking. PARP inhibition decreasing the amount of PARylated proteins associated with SERBP1 and likely all other PARylated proteins is expected.

Evidence from autopsy brain tissue: This study shows unexplained round, punctate staining for SERBP1 in immunohistochemistry (IHC) staining. This may be due to poor preservation of cellular structures in frozen autopsy brain tissue. SERBP1 and pTau co-staining lacks an age matched non-AD control. Most quantifications of human IHC staining and co-localization do not indicate the number of cases and what data points are shown.

The link to stress granules (SGs): G3BP1 staining indicates cytoplasmic mislocalization and perhaps aggregation pathology, but not necessarily SGs. It is not clear whether physiological transient stress granules are preserved in autopsy brain tissue. The co-localization of abundant cytoplasmic G3BP1 and SERBP1 under normal conditions does not indicate association with SGs. Stress granule proteins assemble phase-separated granules in the cytoplasm under cellular stress, whereas here it is shown that normally cytoplasmic SERBP1 has a nucleocytoplasmic distribution in the presence of H2O2, with no evidence for SG formation.

---

## [Author Response]

The following is the authors’ response to the original reviews.

**Reviewer 1 (Public Review):**
Summary:Here the authors convincingly identify and characterize the SERBP1 interactome and further define its role in the nucleus, where it is associated with complexes involved in splicing, cell division, chromosome structure, and ribosome biogenesis. Many of the SERBP1-associated proteins are RNA-binding proteins and SERBP1 exerts its impact, at least in part, through these players. SERBP1 is mostly disordered but along with its associated proteins displays a preference for G4 binding and can bind to PAR and be PARylated. They present data that strongly suggest that complexes in which SERBP1 participates are assembled through G4 or PAR binding. The authors suggest that because SERBP1 lacks traditional functional domains yet is clearly involved in distinct regulatory complexes, SERBP1 likely acts in the early steps of assembly through the recognition of interacting sites present in RNA, DNA, and proteins.Strengths:The data is very convincing and demonstrated through multiple approaches.Weaknesses:No weaknesses were identified by this reviewer.
**Reviewer #2 (Public Review):**
Summary:In this study the authors have used pull-down experiments in a cell line overexpressing tagged SERPINE1 mRNA binding protein 1 (SERBP1) followed by mass spectrometry-based proteomics, to establish its interactome. Extensive analyses are performed to connect the data to published resources. The authors attempt to connect SERBP1 to stress granules and Alzheimer's disease-associated tau pathology. Based on the interactome, the authors propose a cross-talk between SERBP1 and PARP1 functions.Strengths:The main strength of this study lies in the proteomics data analysis, and its effort to connect the data to published studies.Weaknesses:While the authors propose a feedback regulatory model for SERBP1 and PARP1 functions, strong evidence for PARylation modulating SERBP1 functions is lacking. PARP inhibition decreasing the amount of PARylated proteins associated with SERBP1 and likely all other PARylated proteins is expected. This study is also incomplete in its attempt to establish a connection to Alzheimer's disease related tauopathy. A single AD case is not sufficient, and frozen autopsy tissue shows unexplained punctate staining likely due to poor preservation of cellular structures for immunohistochemistry. There is a lack of essential demographic data, source of the tissue, brain regions shown, and whether there was an IRB protocol for the human brain tissue. The presence of phase-separated transient stress granules in an autopsy brain is unlikely, even if G3BP1 staining is present. Normally, stress granule proteins move to the cytoplasm under cellular stress, whereas SERBP1 becomes nuclear. The co-localization of abundant cytoplasmic G3BP1 and SERBP1 under normal conditions does not indicate an association with stress granules.
**Reviewer #3 (Public Review):**
Summary:A survey of SERBP1-associated functions and their impact on the transcriptome upon gene depletion, as well as the identification of chemical inhibitors upon gene over-expression.Strengths:(1) Provides a valuable resource for the community, supported by statistical analyses.(2) Offers a survey of different processes with correlation data, serving as a good starting point for the community to follow up.Weaknesses:(1) The authors provided numerous correlations on diverse topics, from cell division to RNA splicing and PARP1 association, but did not follow up their findings with experiments, offering little mechanistic insight into the actual role of SERBP1. The model in Figure 5D is entirely speculative and lacks data support in the manuscript.

Our article includes several pieces of evidence that support SERBP1’s role in splicing, translation, cell division and association with PARP1. We respectfully disagree that the model in Figure 5D is speculative. The goal of our study was to generate initial evidence of SERBP1 involvement in different biological processes based on its interactome. The characterization of molecular mechanisms in all these scenarios requires a substantial amount work and will the topic of follow up manuscripts.

(2) Following up with experiments to demonstrate that their findings are real (e.g., those related to splicing defects and the PARylation/PAR-binding association) would be beneficial. For example, whether the association between PARP1 and SERBP1 is sensitive to PAR-degrading enzymes is unclear.

We included experiments showing the interaction between endogenous SERBP1 and PARP1. Additionally, we demonstrated that SERBP1 interaction with PARP1 was disrupted when cells are treated with PARP inhibitors.

(3) They did not clearly articulate how experiments were performed. For instance, the drug screen and even the initial experiment involving the pull-down were poorly described. Many in the community may not be familiar with vectors such as pSBP or pUltra without looking up details.

We provided additional details about the vectors and expanded the description of experiments in results and figure legends.

(4) The co-staining of SERBP1 with pTau, PARP1, and G3BP1 in the brain is interesting, but it would be beneficial to follow up with immunoprecipitation in normal and patient samples to confirm the increased physical association.

Thank you for this suggestion. We performed instead a Proximity Ligation Assay (PLA) on human tissue. Data was included in Fig. 7B and C. PLA between pTau and SERBP1 confirmed interaction in AD cortices as well as SERBP1 with PARP1.

(5) The combination index of 0.7-0.9 for PJ34 + siSERBP1 is weak. Could this be due to the non-specific nature of the drug against other PARPs? Have the authors looked into this possibility?

The combination index could be considered weak in the case of U251 cells but not in the case of U343 cells. PJ34 has been shown to be mainly a PARP-1 inhibitor. Different PJ34 concentrations and different drugs will be examined in future studies. It is worth mentioning that in a genetic screening, SERBP1 has been shown to increase sensitivity to different PARP inhibitors (PMID: 37160887). This information is included in the manuscript.

**Reviewer #1 (Recommendations For The Authors):**
This is a really well-done piece of research that is written very well. The data are very convincing and the conclusions are well supported. Some wording in Figures 2B and D is pixelated and hard to read. All the figure legends could benefit from being expanded but this is especially true for Figures 2, 3, 7, and 8. There is a ton of data being presented and a very limited description of what was done and what is being concluded. Some of the content may not be fully comprehended by some readers with limited descriptions.

We revised all figures to assure images are clear and their resolution is high. We expanded all figure legends to provide a better explanation of the experimental design.

**Reviewer #2 (Recommendations For The Authors):**
The "merged" pdf file is the same as the "article".

Individual files were uploaded this time.

The abstract should spell out acronyms, such as the name of the protein Serpine1 mRNA-binding protein 1 (SERBP1).

This was not included since the abstract has a word limit.

"SERBP1 (Serpine1 mRNA-binding protein 1) is a unique member of this group of RBPs". In what way is it unique?

The text was modified to better explain SERBP1’s singularities.

"RBPs containing IDRs and RGG motifs are particularly relevant in the nervous system. Their misfolding contributes to the formation of pathological protein aggregates in Alzheimer's disease (AD), Frontotemporal Lobar Dementia (FTLD), Amyotrophic Lateral Sclerosis (ALS), and Parkinson's disease (PD)" -> while TDP-43 and FUS in ALS/FTD may fit this description, it is not true for tau and amyloid-beta (AD) and alpha-synuclein (PD).

"SERBP1 is a unique RBPs containing IDRs and RGG motifs yet lacks other readily recognizable, canonical or structured RNA binding motifs. Moreover, SERBP1 has been observed by our study and others as common Tau interactor in Alzheimer’s Disease (AD) brains. RBPs containing IDRs (e.g. TDP-43, FUS, hnRNPs, TIA1) have been shown self-aggregate and co-aggregate with pathogenic amyloids (Tau, Aβ-amyloid and α-Synuclein) in AD, Frontotemporal Lobar Dementia (FTLD), Amyotrophic Lateral Sclerosis (ALS), and Parkinson's disease (PD) and this suggest that, like other IDRs RBPs, SERBP1 contributes to RNA dysmetabolism in neurodegenerative diseases”.

While the authors propose a feedback regulatory model for SERBP1 and PARP1 functions, strong evidence for PARylation modulating SERBP1 functions is lacking. The fact that PARP inhibition decreases the amount of PARylated proteins associated with SERBP1 and likely all other PARylated proteins is expected and cannot count as evidence.

We included data showing that treatment with PJ34 (PARP inhibitor) decreases SERBP1 interaction with PARP1 and G3BP1. We are currently conducting a more extensive analysis to identify SERBP1 PAR binding domain and the impact of PARP inhibition on its interactions and functions. These experiments will be included in a new manuscript.

A single AD case is not sufficient.

Sorry for the poor clarity. We included in the study 6 cases from age-matched controls and 6 cases of AD. We summarize all cases demographics, and the experimental application assigned to each case in Table 1. Moreover, we included a paragraph regarding Human tissue harvesting.

Most western blot data are not quantified from multiple replicates, as required.

Quantifications are now provided.

FTLD - frontotemporal lobar degeneration (not dementia).

This was corrected.

Frozen autopsy tissue is problematic due to poor preservation. The staining presented here shows unexplained punctate staining likely due to poor preservation of cellular structures for immunohistochemistry.

We included a paragraph regarding human tissue harvesting. We have successfully used frozen tissues in our previous studies, observing a well preserved neuronal and tissue structure (PMIDs: 32855391, 31532069 and 30367664)

The presence of phase-separated stress granules in tissue is controversial since these are transient structures.Normally, stress granule proteins move to the cytoplasm under cellular stress, whereas SERBP1 becomes nuclear. The co-localization of abundant (and partially overexposed) cytoplasmic G3BP1 and SERBP1 under normal conditions is not evidence for association with stress granules. Does induction of stress granule formation lead to colocalization in stress granules? The H2O2 experiment suggests otherwise.

RBPs implicated in stress response move to stress granules when cells are exposed to stress. SERBP1 has been shown to shuttle to stress granules and nucleus in stress conditions (PMID: 24205981). Our results are in agreement.

Using co-IF, we observed some overlap between G3BP1 and SERBP1 in AD tissues. As shown in Fig. S6A and B, 50% of stress granules overlap with SERBP1 signal. On the contrary, it is hard to assess their relationship in aged-matched control brains where stress granules form and accumulate with a lower rate than in AD. SERBP1 is not very abundant in normal brains. It is known that RNA-Binding Proteins aggregation and/or dysfunctional LLPS dysregulate stress granules formation and accumulation in AD and other proteinopathies (PMIDs 30853299, 27256390 and 31911437). However, it is too early to determine the role of SERBP1 and its contribution to stress granules formation and accumulation. We will examine this topic in future studies.

There is a lack of essential demographics data (age, clinical diagnosis, path diagnosis, co-pathologies, Braak stage, etc.), source of the tissue (what brain bank?), brain regions shown, and whether there was informed consent for the collection and use of human brain tissue.

We included the information requested in materials and methods section.

**Reviewer #3 (Recommendations For The Authors):**
The authors need to better explain their experimental rationale and approach in the main text, not just in the supplementary materials.

We have extensively revised the text to provide a better description of experiments in the results section and figure legends.